# Study on the Alleviating Effect and Potential Mechanism of Ethanolic Extract of *Limonium aureum* (L.) Hill. on Lipopolysaccharide-Induced Inflammatory Responses in Macrophages

**DOI:** 10.3390/ijms242216272

**Published:** 2023-11-13

**Authors:** Zhen Yang, Jingyuan Man, Yu Liu, Hongjuan Zhang, Di Wu, Dan Shao, Baocheng Hao, Shengyi Wang

**Affiliations:** 1Key Laboratory of New Animal Drug Project, Key Laboratory of Veterinary Pharmaceutical Development, Ministry of Agriculture and Rural Affairs, Lanzhou Institute of Husbandry and Pharmaceutical Sciences, Chinese Academy of Agriculture Sciences, Lanzhou 730050, China; lzmy_yz@163.com (Z.Y.); liuyu8108@163.com (Y.L.); zhanghongjuan@caas.cn (H.Z.); wudi@caas.cn (D.W.); shaodan@caas.cn (D.S.); 2College of Veterinary Medicine, Gansu Agricultural University, Lanzhou 730070, China; m1013264544@gmail.com

**Keywords:** *Limonium aureum*, inflammation, macrophage, AKT1/RELA/PTGS2 signaling pathway, MAPK3/JUN signaling pathway

## Abstract

Inflammation is the host response of immune cells during infection and traumatic tissue injury. An uncontrolled inflammatory response leads to inflammatory cascade, which in turn triggers a variety of diseases threatening human and animal health. The use of existing inflammatory therapeutic drugs is constrained by their high cost and susceptibility to systemic side effects, and therefore new therapeutic candidates for inflammatory diseases need to be urgently developed. Natural products are characterized by wide sources and rich pharmacological activities, which are valuable resources for the development of new drugs. This study aimed to uncover the alleviating effect and potential mechanism of natural product *Limonium aureum* (LAH) on LPS-induced inflammatory responses in macrophages. The experimental results showed that the optimized conditions for LAH ultrasound-assisted extraction via response surface methodology were an ethanol concentration of 72%, a material-to-solvent ratio of 1:37 g/mL, an extraction temperature of 73 °C, and an extraction power of 70 W, and the average extraction rate of LAH total flavonoids was 0.3776%. Then, data of 1666 components in LAH ethanol extracts were obtained through quasi-targeted metabolomics analysis. The ELISA showed that LAH significantly inhibited the production of pro-inflammatory cytokines while promoting the secretion of anti-inflammatory cytokines. Finally, combined with the results of network pharmacology analysis and protein expression validation of hub genes, it was speculated that LAH may alleviate LPS-induced inflammatory responses of macrophages through the AKT1/RELA/PTGS2 signaling pathway and the MAPK3/JUN signaling pathway. This study preliminarily revealed the anti-inflammatory activity of LAH and the molecular mechanism of its anti-inflammatory action, and provided a theoretical basis for the development of LAH as a new natural anti-inflammatory drug.

## 1. Introduction

Inflammation is a host response by immune cells due to disruption of homeostasis in vivo during infection and traumatic tissue injury. This complex biological process plays a critical role in the health and survival of multicellular organisms by mediating host pathogen defense, tissue repair, and restoration of homeostasis in vivo [1,2]. The controlled inflammatory responses can clear harmful stimuli and restore normal physiological functions of the body, while the opposite leads to the occurrence of disease. Inflammatory factors, including biological, chemical, physical, and other factors, can trigger inflammatory responses. Inflammation can be categorized into acute inflammation and chronic inflammation, according to the type and duration of the stimulus, and they sustain a long-term coexistence [2,3]. A failure of the immune system to eliminate the destructive effects of acute inflammation in a timely manner results in an inflammatory cascade, thereby translating into chronic inflammation, which leads to the occurrence and development of a variety of diseases, such as chronic asthma, cardiovascular disease, diabetes, cancer, etc. [4,5,6]. Lipopolysaccharide (LPS), an endotoxin, is a chemical component characteristic of the outer wall layer of the cell wall of Gram-negative bacteria, and consists of lipids and polysaccharides [7]. Studies have shown that LPS plays an important role in inflammatory response by activating Toll-like receptor 4 (TLR4), which is present on the cell membrane surface of host cells, to induce the production of a large number of cytokines including tumor necrosis factor-alpha (TNF-α), interleukin-6 (IL-6), interleukin-8 (IL-8), inducible nitric oxide synthase (iNOS), etc. [8,9]. LPS, as an effective cytotoxic inducer of inflammatory response, and its induced inflammation model have been widely recognized and applied in the development of anti-inflammatory drugs [10,11].

Macrophages are the key components of innate immunity and play a critical role in inflammatory and tissue repair responses. As the body’s first line of defense against pathogens, macrophages exhibit a high degree of diversity and plasticity when induced by different stimuli, and thus perform multiple functions [12,13]. Macrophages stimulated by different microenvironments can be polarized into the “classically activated” M1 type and “alternatively activated” M2 type. These two phenotypes of macrophages play completely different roles in the pathophysiological process of the body [14]. Under the stimulation of T helper cytokine 1 (Th1) such as LPS and interferon-γ (IFN-γ), macrophages exhibit the M1 phenotype, participate in the early inflammatory immune response, and produce a large number of inflammatory mediators [15,16]. Macrophages activated by Th2 cytokines (IL-13, IL-14) and macrophage colony-stimulating factor (M-CSF) exhibit an M2 phenotype, produce anti-inflammatory cytokines, and contribute to tissue repair responses [17]. Studies have demonstrated that the interconverting nature of macrophage phenotypes is largely responsible for their dual role in disease processes [18]. In view of the important role of macrophages in inflammatory response, the LPS-stimulated macrophage inflammation model has become a well-established in vitro model to study the effects and mechanisms of anti-inflammatory drug candidates.

Currently, the main therapeutic drugs used against inflammation are corticosteroids, nonsteroidal anti-inflammatory drugs (NSAIDs), and biologics agents [19]. In addition to the price issue, the systemic side effects associated with their long-term use, such as osteoporosis, cardiovascular disease, renal disease, immunosuppression, and hepatotoxicity, are also important reasons that limit their use [20]. Natural products are secondary metabolites synthesized in living organisms and have been used in the prevention and treatment of diseases since ancient times [21]. In recent years, with continuous progress in drug screening and development technology, natural products have become one of the valuable resources for the development of new drugs due to their wide sources and rich pharmacological active ingredients [22]. Flavonoids are a class of secondary metabolites synthesized through flavonoid and phenylpropanoid metabolic pathways with a parent nucleus of 2-phenylchromone, that are widely present in fruits, vegetables, grains, tea, as well as on the plant epidermis, flowers, rhizomes, and other parts of the plant [23]. It has been reported that flavonoids can block the activation cascade of inflammatory signaling pathways by reducing the phosphorylation or nuclear transcription of inflammation-related genes, thereby inhibiting the release of inflammatory mediators such as pro-inflammatory cytokines to play a role in alleviating inflammation [24,25]. With in-depth research on the anti-inflammatory activities and mechanisms of flavonoids, more and more naturally derived flavonoids and their derivatives have emerged as new drug candidates for the prevention and treatment of inflammatory diseases.

*Limonium aureum* (L.) Hill. (here after referred to as LAH) is a perennial herb in the genus *Limonium* Mill. of the *Plumbaginaceae*; it is a xerophytic salt-secreting plant with strong salt tolerance and widely distributed in northern China. Given the anti-inflammatory activity of flavonoids, LAH is considered an excellent natural medicinal product as it contains a variety of flavonoids, such as homoeriodectyol, eriodectyol, naringenin, myricetin 3-O-β-D-glucopyranoside, myricetin 3-O-β-D-galactoside, and quercetin [26]. However, most of the current studies on LAH have focused on the extraction and isolation of its active ingredients, structural identification, and tissue culture, while few studies have been carried out on its anti-inflammatory activity and related mechanism of action. Therefore, the present study was conducted to investigate the anti-inflammatory effects and potential mechanisms of LAH on the LPS-induced inflammatory macrophage model by integrating the application of network pharmacology, molecular biology, and other research methods. Firstly, the process of ultrasound-assisted extraction of LAH was optimized using response surface methodology (RSM). Then, the LPS-induced inflammation model of RAW264.7 cells was established, and the effects of LAH on the secretion of the cellular pro-inflammatory cytokines and anti-inflammatory cytokines were detected using an ELISA to evaluate the protective effects of LAH on the cellular inflammatory response. Next, the metabolites in LAH extracts were characterized and quantified via metabolomics study and analyzed via network pharmacology. Finally, the results of the network pharmacology analysis were validated by detecting the protein expression levels of the anti-inflammatory targets of LAH, and the potential anti-inflammatory mechanism of action of LAH was speculated according to the available data. Collectively, this study lays the foundation for further elucidating the anti-inflammatory activity and potential mechanism of LAH, and promotes its development and application as a natural therapeutic drug for inflammation.

## 2. Results

### 2.1. Optimization of Extraction Technology of Total Flavonoids from LAH Using Response Surface Methodology

With a wide range of biological activities and fewer side effects, natural products have been useful in the treatment of a variety of diseases including inflammation, tumors, metabolic diseases, etc., and are the most well-known database for the discovery of new drugs [27,28,29]. LAH is a natural medicinal plant with good bioactivity, but there are few studies on its active ingredients and bioactivity. In view of the advantages of UAE, such as simple operation, low cost, and high extraction efficiency, UAE technology was used in this study for the extraction of LAH. In addition, during the extraction of natural products, changes in factors such as extraction time, extraction temperature, material-to-solvent ratio, and solvent type can lead to changes in the yield of the product to be extracted. On the other hand, the optimization of parameters affecting the extraction process of natural products using experimental design has significant advantages in improving the extraction efficiency, data quality, and stability of the extraction rate [30]. Therefore, the response surface method was further used to optimize the four parameters of ethanol concentration, material-to-solvent ratio, extraction temperature, and extraction power.

#### 2.1.1. Plotting of Standard Curve

The standard curve of quercetin is shown in Figure 1A. The linear regression equation is as follows:Y = 19.64x + 0.0565 (R^2^ = 0.9992)

It was demonstrated that quercetin had a good linear relationship in the concentration range of 0.0025–0.0125 mg/mL. The extraction rate of LAH was indexed by the extraction rate of total flavonoids of the extract, and the formula for the extraction rate of total flavonoids of LAH was as follows:Extraction rate of total flavonoids of LAH = N·C·V/W
where N represents the number of dilutions, C is the measured concentration of total flavonoids of LAH, V is the volume of the sample solution at the time of extraction and reconstitution, and W is the weight of the sample of the dried powder of LAH at the time of extraction.

#### 2.1.2. Single-Factor Experiment

The results of the single-factor experiment affecting the extraction rate of LAH are shown in Figure 1. Under the premise of only one variable, when the concentration of ethanol increased, the extraction of LAH first increased and then decreased, reaching a peak at an ethanol concentration of 70%, so it was more appropriate to choose 60–80% of the concentration of ethanol in the extraction solvent for the follow-up study. When the material-to-solvent ratio was increased to 1:40 g/mL, the highest extraction rate of LAH was obtained, and then it showed a decreasing trend, so a material-to-solvent ratio range of 1:30–1:50 g/mL was selected as appropriate. When the extraction temperature was 70 °C, the maximum extraction rate was reached, so an extraction temperature of 60–80 °C was chosen for the subsequent study. When the ultrasonic power increased, the extraction of LAH first increased, reaching a peak at 70 W, and then showed a decreasing trend, so an extraction power range of 60–80 W was selected.

#### 2.1.3. Optimization of LAH Extraction Process Using Response Surface Methodology

The experimental program obtained by Design-Expert 8.0.6 and the corresponding results are illustrated in Appendix A. The quadratic multiple regression equations obtained from the subsequent analysis of the extraction rate of total flavonoids of LAH in relation to the ethanol concentration, material-to-solvent ratio, extraction temperature, and extraction power were obtained as follows:Y = 0.37 + 3.958·10^−3^·A − 0.02B + 0.014C + 6.867·10^−3^·D − 0.017AB + 1.925·10^−3^·AC + 0.011AD − 5.325·10^−3^·BC + 0.013BD + 3.825·10^−3^·CD − 0.031A^2^ − 0.035B^2^ − 0.025C^2^ − 0.023D^2^
where Y is the extraction rate of total flavonoids of LAH, A is the ethanol concentration, B is the material-to-solvent ratio, C is the extraction temperature, and D is the extraction power. The variance analysis data of the constructed response surface model are shown in Table 1; the data had statistical significance (*p* < 0.05) and lower error (*p* > 0.05 for lack of fit), indicating that the model was successfully constructed and could be used to analyze the ultrasonication-assisted extraction process of LAH. According to the F-value, the factors that had the greatest influence on the extraction rate of total flavonoids of LAH were the material-to-solvent ratio, which had a significant effect (*p* < 0.05), followed by the ultrasonic temperature and power, while the ethanol concentration had the least influence.

The response surface models and pairwise interaction contour plots of the effect of different factors on the extraction rate of LAH total flavonoids are shown in Figure 2. It can be seen that the interaction of the three factors, ethanol concentration, extraction temperature, and extraction power, had a non-significant effect on the extraction rate of LAH total flavonoids, and the effect of the extraction temperature was greater than the effect of the extraction power, which, in turn, was greater than the effect of the ethanol concentration. With an increase in these variables, the extraction rate of LAH total flavonoids first increased and then decreased. The contour lines were denser and the variation of the extraction rate of LAH total flavonoids was larger when the extraction power was in the range of 70–80 W and the extraction temperature was in the range of 70–80 °C. It can be seen that the difference of its interaction with the remaining three factors on the extraction rate of LAH total flavonoids was very significant when the material-to-solvent ratio was the variable. When the material-to-solvent ratio was 1:35–1:45, the contour was very dense, indicating that it had the greatest effect on the extraction rate of LAH total flavonoids.

#### 2.1.4. Verification Test

The optimal process conditions for LAH extraction were analyzed using the Box–Behnken software; they were an ethanol concentration of 71.85%, a material-to-solvent ratio of 1:36.73 g/mL, an extraction temperature of 73.18 °C, and an extraction power of 71.32 W. The predicted extraction rate of LAH total flavonoids was 0.3804%. The extraction conditions were changed to an ethanol concentration of 72%, a material-to-solvent ratio of 1:37 g/mL, an extraction temperature of 73 °C, and an extraction power of 70 W, according to the actual operation situation. After the actual operation, the extraction rates of LAH total flavonoids were measured as shown in Table 2. The average extraction rate of LAH total flavonoids was 0.3776%, which had little difference with the predicted value of the model, and the RSD was 3.277%, which was less than 5%, indicating that the optimal extraction conditions analyzed using the response surface method were accurate and reliable.

In this study, the UAE technique was selected with its characteristics of simple operation, low cost, and high extraction efficiency to extract the natural product LAH, and the optimal extraction process was obtained using the response surface method, which lays the foundation for the subsequent study of the biological activity of LAH. From the results of the single-factor experiment it can be seen that, with the increase in ethanol concentration, temperature, power, and material-to-solvent ratio, the total flavonoid extraction of LAH first rose and then fell. When the factors interacted with each other, the extraction rate of total flavonoids in LAH also increased first and then decreased. This indicates that the effects of these four factors on the extraction rate of LAH total flavonoids were not in the same trend, and reaching a certain limit and then rising will lead to a decrease in the yield of LAH total flavonoids. The pairwise interaction of different factors on the response value can be judged by the inclination degree of the response surface and the density and shape of the contours. The greater the impact of the interaction between the two on the response value, the more inclined the response surface is; conversely, the smaller the inclination degree of the response surface, the lower the degree of interaction [31]. As shown in Figure 2, the inclination degree of the response surface of the interaction of the four factors was low, indicating that the interaction of the four factors did not have a significant effect on the extraction rate of total flavonoids of LAH.

Guided by the parameters optimized using the response surface method, the optimal process conditions for the ultrasonic-assisted extraction of LAH were finally determined to be an ethanol concentration of 72%, a material-to-solvent ratio of 1:37 g/mL, an extraction temperature of 73 °C, and an extraction power of 70 W, according to the actual experimental operating conditions. It was determined that the actual average extraction rate of total flavonoids of LAH under this condition was 0.3776%, which was close to the predicted value. However, in addition to the four factors selected in this paper, the extraction frequency, extraction method, and extraction solvent also have impacts on the extraction rate of LAH. Therefore, other factors affecting the extraction rate of LAH can be further considered in subsequent studies.

### 2.2. Quasi-Targeted Metabolomics Analysis

In recent years, research on the active ingredients and biological activities of LAH has stagnated, and there are still many unanswered questions. Quasi-targeted metabolomics is a novel metabolomics assay that combines the advantages of targeted and untargeted metabolomics with high throughput, high accuracy, and high sensitivity. It is based on the SCIEX QTRAP^®^ 6500+ mass spectrometer with a triple quadrupole-linear ion trap composite, and employs the multiple reaction monitoring (MRM) mode for the characterization and quantification of metabolites in samples [32]. This study used quasi-targeted metabolomics technology to analyze the ingredient information of the LAH extract, aiming to provide a reference for revealing the pharmacological activity of LAH, which has positive significance for the utilization of the medicinal value of natural products.

The LAH extracts were assayed in multiple reaction monitoring mode (MRM), based on the database constructed by Novogene Co., Ltd. (Beijing, China), in which the characterization of metabolites was analyzed based on Q1/Q3 (ion pair), RT (retention time), DP (de-clustering potential), and CE (collision energy) for each compound. And the quantification of metabolites was analyzed based on the peak area of Q3 (daughter ion) using the MRM mode of triple quadrupole. Briefly, the characteristic ions of each substance were screened out by the triple quadrupole, and the signal intensity of the characteristic ions (CPS) was obtained in the detector. The downstream mass spectrometry files of the samples were analyzed with the SCIEX OS V1.4 software to perform the integration and correction of the chromatographic peaks, and the peak area (Area) of each chromatographic peak represented the relative quantitative value of the corresponding substance. Through quasi-targeted metabolomics analysis, a total of 1666 qualitative and quantitative data of components in LAH extracts were finally obtained, as shown in Appendix A.

In this study, a total of 1666 components in LAH extracts were identified, and their relative quantitative values were obtained separately via quasi-targeted metabolomics analysis. This is of positive significance for the further understanding of the pharmacologically active components in LAH. However, our study still has some limitations. These include the fact that it is unclear which specific components of the LAH extract mixture exert pharmacological activity. Although quasi-targeted metabolomics analyses provided some information on the components contained in LAH, more in-depth studies are needed to determine which compounds are specifically responsible for the pharmacological activity of LAH.

### 2.3. The Effects of LAH on the Viability of RAW264.7 Cells

A CCK-8 analysis was used to determine the effects of different concentrations of LAH on cell viability in this study, which reflects the cell proliferation activity by detecting the concentration of succinate dehydrogenase in mitochondria in cells. By determining the effect of different concentrations of LAH on cell viability, the drug dosages without cytotoxicity can be effectively screened for further studies.

As shown in Figure 3A, the cell viability tended to increase and then decrease with the increase in the concentration of LAH. The cell viability peaked when the concentration of LAH was 30 µg/mL, while the cell viability decreased significantly when the concentration of LAH was greater than 100 µg/mL. In addition, compared with the blank control group, the cell viability increased significantly at LAH concentrations in the range of 2–100 µg/mL (*p* < 0.01), suggesting that LAH had a certain promotion effect on cell proliferation in this concentration range. Therefore, 10, 30, and 50 µg/mL were selected as low, medium, and high dose groups of LAH for subsequent experiments.

Cell proliferation is one of the fundamental processes in the development of organisms and the continuation of life, and its regulation in organisms is influenced by a variety of factors and mechanisms, including growth factors, cell cycle regulation, apoptosis regulation, and oxidative metabolism regulation. The results of our experiment indicated that LAH concentrations below 100 µg/mL are non-cytotoxic, and the selection of subsequent experimental doses in the range of LAH concentrations below this value will facilitate the study of the pharmacological activity of LAH. Furthermore, as can be seen in Figure 3A, 10–50 µg/mL of LAH resulted in a significant increase in cell activity, and 30 µg/mL of LAH increased the activity of RAW264.7 cells by more than 150%. Since the main objective of this study was to investigate the anti-inflammatory activity and related mechanism of LAH, a more detailed experimental design is needed to account for the role of LAH in promoting cell proliferation.

### 2.4. The Effects of LAH on the Secretion of the Cellular Pro-Inflammatory Cytokines and Anti-Inflammatory Cytokines

Cytokines are a large class of multifunctional, small-molecular-weight, secreted proteins. They regulate the immune response through a variety of mechanisms and thus play a key role in the activation of pro- and anti-inflammatory pathways in the innate and acquired immune system [33]. Many pro-inflammatory and anti-inflammatory cytokines are involved in the body’s inflammatory response, and both are activated simultaneously during the inflammatory response [34]. Cytokines serve as sensitive indicators of inflammation, and monitoring their levels can determine the extent of disease progression [33].

For the purpose of understand the anti-inflammatory activity of LAH, the effects of LAH on the secretion of the cellular pro-inflammatory cytokines and anti-inflammatory cytokines in LPS-induced RAW264.7 cells were evaluated. As shown in Figure 3B,C, LPS exposure significantly decreased the concentrations of anti-inflammatory cytokines IL-4 and IL-10 in the supernatants of the cell culture medium (*p* < 0.01), whereas the effects of LPS on anti-inflammatory cytokines were differently reversed in each dose group of LAH in a dose-dependent trend. Meanwhile, LPS induced cells to secrete large amounts of the pro-inflammatory cytokines IFN-γ, IL-6, and TNF-α (Figure 3D–F), which resulted in significantly higher concentrations of pro-inflammatory cytokines in the LPS group than in the blank control group (*p* < 0.01). In contrast, after treatment with different doses of LAH, the release of IFN-γ, IL-6, and TNF-α was reduced to different degrees, with the differences between the medium- and high-dose groups compared with the LPS group being significant (*p* < 0.05), and again with a certain dose-dependent trend. The data obtained suggested that LAH may exert its anti-inflammatory effects by promoting the secretion of anti-inflammatory cytokines and inhibiting the production of pro-inflammatory cytokines.

In this study, the secretion levels of pro- and anti-inflammatory cytokines in macrophages under different treatment conditions were detected using ELISA, thus demonstrating the in vitro anti-inflammatory effect of LAH. Of course, in addition to detecting the effect of LAH on the secretion level of cytokines, the detection of mRNA and protein levels can further confirm the alleviating effect of LAH on LPS-induced inflammation in macrophages. Moreover, based on the in vitro studies, the in vivo anti-inflammatory effects of LAH still need to be examined in the future, so as to better lay the theoretical research foundation for developing LAH into a natural anti-inflammatory drug.

### 2.5. Network Pharmacology Analysis

Network pharmacology is an emerging discipline that integrates multidisciplinary contents with systems biology as the basic theory, and reveals the relationship between drugs and diseases by constructing the network of “disease-gene-target-drug” [35]. It typically uses information on partial correlations, conditional interactions, or computerized learning methods along with bulk or single-cell transcriptomics and other omics data to identify associations between genes [36]. The pharmacological network inferred by this method helps to analyze the genes susceptible to disruption in the disease signaling pathways, leading to the speculation of drug targets, and further validation of hypotheses can be conducted through experiments [37]. Therefore, the present study used a network pharmacology approach to analyze the anti-inflammatory mechanism and hub genes of LAH from several aspects, such as candidate component analysis, target disease database construction, hub gene prediction, network analysis, and enrichment analysis.

#### 2.5.1. Screening for Active Ingredients of LAH

The high-throughput traditional Chinese medicine databases TCMSP (https://old.tcmsp-e.com/tcmsp.php, accessed on 17 May 2022), ETCM (http://www.tcmip.cn/ETCM/index.php, accessed on 17 May 2022), and Herb (http://herb.ac.cn/, accessed on 17 May 2022) were used to query the information of 1666 substances in LAH extract identified in the quasi-targeted metabolomics analysis, and a total of 74 substances with OB > 30%, DL > 0.18, and relative content > 1 × 10^6^ were screened. In addition, there were another 22 components with OB < 30% or DL < 0.18, but relative content > 1 × 10^6^, whose anti-inflammatory effects were reported in the literature. Therefore, a total of 96 active ingredients in LAH were screened. The targets of action of each active ingredient were queried through the SwissTarget Prediction database and subsequently analyzed, and the screening results of the active ingredients of the LAH extract are shown in Appendix A.

#### 2.5.2. Construction of Common Targets between Active Ingredients of LAH and Diseases

A total of 396 targets were identified by querying and de-emphasizing the 96 active ingredients of LAH through the SwissTarget Prediction database. The Genecards database was searched with the keyword “anti-inflammatory”, and 3695 anti-inflammatory-related targets were found. Taking the intersection of LAH active ingredient targets and anti-inflammatory targets, 255 targets related to the anti-inflammatory effect of LAH were obtained, as shown in Figure 4A.

#### 2.5.3. Analyses of GO and KEGG Pathway Enrichment

The results of the GO enrichment analysis are shown in Figure 4B. The LAH anti-inflammatory biological processes (BP) mainly involved signal transduction, positive regulation of transcription from RNA polymerase II promoter, response to drugs, G-protein-coupled receptor signaling pathway, negative regulation of transcription from RNA polymerase II promoter, negative regulation of apoptotic process, response to xenobiotic stimulus, inflammatory response, positive regulation of cell proliferation, and positive regulation of transcription, DNA-templated. The top 10 significant enrichment terms for cellular components (CC) included plasma membrane, integral component of membrane, cytoplasm, nucleus, nucleoplasm, integral component of plasma membrane, extracellular exosome, membrane, and extracellular region. The molecular function (MF) enrichment results indicated that most of these targets were associated with protein binding, identical protein binding, enzyme binding, zinc ion binding, ATP binding, RNA polymerase II core promoter proximal region sequence-specific DNA binding, protein homodimerization activity, RNA polymerase II transcription factor activity, sequence-specific DNA binding, DNA binding, and transcription factor activity. In addition, the KEGG pathway enrichment analysis (Figure 4C) showed that KEGG pathways based on LAH anti-inflammatory targets were mainly associated with pathways in cancer, cAMP signaling pathway, HIF-1 signaling pathway, NF-κB signaling pathway, PI3K-A KT signaling pathway, JAK-STAT signaling pathway, B cell receptor signaling pathway, Toll-like receptor signaling pathway, MAPK signaling pathway, and mTOR signaling pathway.

#### 2.5.4. Construction of Protein–Protein Interaction Network

To elucidate the potential anti-inflammatory mechanism of LAH, the 255 cross-targets obtained above were subjected to protein–protein interaction analysis using the String database, and a medium-confidence interaction score was set to construct the PPI network. The results of protein–protein interaction analysis were visualized using Cytoscape software, as shown in Figure 4D. Potential targets were represented by nodes, and the size and color of the nodes were adjusted according to the degree value, which became larger and darker as the degree value increased. The interactions between targets were represented by edges, and the line thickness was used to indicate the combined score of interactions between targets. The results of the PPI network analysis are shown in Appendix A, and the combined scores between nodes ranged from 0.4 to 0.999. Finally, the hub genes were screened for validation based on the degree values.

After a comprehensive network pharmacology analysis, a total of 396 targets of 96 active ingredients in LAH were screened, among which 255 targets were related to anti-inflammatory effects. Through the topological characterization of the PPI network constructed by these 255 anti-inflammatory targets, hub genes including AKT1, MAPK3, RELA, PTGS2, and JUN were screened. In addition, the biological processes, cellular components, molecular functions, and signaling pathways involved in the anti-inflammatory effects of LAH were obtained using GO enrichment analysis and KEEG pathway analysis.

### 2.6. Protein Expression Levels of Anti-Inflammatory-Related Targets of LAH

Western blotting is a routine technique for protein analysis, which is a method to detect the expression of a certain protein in a sample based on the specific binding of antigen and antibody [38]. Because of the advantages it shows in the detection of target proteins with high sensitivity and specificity, Western blotting has become an essential experimental technique in the study of drug mechanisms of action.

In order to investigate the mechanism by which LAH exerts anti-inflammatory effects on LPS-induced inflammatory macrophages, the protein expression levels of the anti-inflammatory hub genes of LAH screened using network pharmacology were determined using the Western blotting technique. The results showed that the protein expression levels of AKT1, PTGS2, RELA, JUN, and MAPK3 were significantly up-regulated in RAW264.7 cells from the LPS-treated group compared with the blank control group (*p* < 0.05, Figure 5A–F), whereas different concentrations of LAH significantly down-regulated the protein expression levels of AKT1, PTGS2, RELA, JUN, and MAPK3 in LPS-induced RAW264.7 cells as compared with the LPS-treated group (*p* < 0.05, Figure 5A–F). The above results suggested that LAH may exert its protective effect on LPS-induced inflammatory macrophages by regulating hub genes including AKT1, PTGS2, RELA, JUN, and MAPK3.

In this study, the results of the network pharmacological analysis were verified by examining the protein expression levels of the anti-inflammatory hub genes of LAH. Based on the protein expression of the hub genes, it is known that LAH may alleviate the inflammatory response of macrophages by inhibiting the LPS-induced up-regulation of the protein expression levels of AKT1, PTGS2, RELA, JUN, and MAPK3. In addition, the use of selective activators/inhibitors and fluorescent labeling probes in future experimental design will help to further reveal the interactions among hub genes in order to better elucidate the mechanism of action of LAH in alleviating macrophage inflammation.

## 3. Materials and Methods

### 3.1. Materials and Reagents

Whole plants of the *Limonium aureum* (L.) Hill. species were collected in August 2022 at the Dawa Mountain Comprehensive Experimental Base of the Lanzhou Institute of Husbandry and Pharmaceutical Sciences of CAAS (Lanzhou, China). The sample was identified by the Prataculture research group of the Lanzhou Institute of Husbandry and Pharmaceutical Sciences of CAAS (Lanzhou, China). A Cell Counting Kit-8 (CCK-8) was purchased from Biosharp Life Sciences (Hefei, China). Fetal bovine serum (FBS) and Dulbecco’s modified Eagle’s medium (DMEM) high glucose were obtained from Gibcol Life Technology (New York, NY, USA) and HyClone (Logan, UT, USA). ELISA kits were obtained from Jiangsu Meimian Industrial Co., Ltd., (Yancheng, China). Murine macrophage cell line RAW264.7 cells were provided by the Cell Culture Center of the Chinese Academy of Sciences (Shanghai, China). Anti-PTGS2 (Cat# 66351-1-Ig), Anti-NF-κB p65 (Cat# 80979-1-RR), Anti-AKT (Cat# 60203-2-Ig), Anti-ERK1/2 (Cat# 11257-1-AP), and Anti-JUN (Cat# 66313-1-Ig) bodies were purchased from Proteintech Group, Inc. (Rosemont, IL, USA). Lipopolysaccharides (LPS) from *Escherichia coli* O55:B5 was purchased from Sigma-Aldrich (Shanghai, China) and Trading Co., Ltd., (Shanghai, China).

### 3.2. Optimization of Extraction Technology of Total Flavonoids from LAH Using Response Surface Methodology

#### 3.2.1. Preparation of Dry Powder of LAH

LAH was harvested at the Dawa Mountain Comprehensive Experimental Base of the Lanzhou Institute of Husbandry and Pharmaceutical Sciences of CAAS (Lanzhou, China). The fresh plants were air-dried at room temperature, avoiding light for one week. Then, the dried plants were ground to powder for later use.

#### 3.2.2. Plotting of Standard Curve

Quercetin standard was accurately weighed and dissolved in 70% (*v*/*v*) ethanol solution to obtain the quercetin standard solutions with the concentrations of 0.0025, 0.005, 0.0075, 0.01, and 0.0125 mg/mL, respectively. The absorbance of 370 nm was determined with 70% ethanol as the blank control. The standard curve was plotted with the concentration C (mg/mL) as the horizontal coordinate and the absorbance A as the vertical coordinate.

#### 3.2.3. Single-Factor Experiment

The effects of four factors, including ethanol concentration in solvent, material-to-liquid ratio, extraction temperature, and extraction power, on the yield of total flavonoids in 1 g of LAH extract were investigated using the controlled variable method with the yield of total flavonoids as the index. Six levels were set for each factor; to be specific, the concentration of ethanol in the solvent water was set to 30%, 40%, 50%, 60%, 70%, and 80%, the material-to-liquid ratio was set to 1:10, 1:20, 1:30, 1:40, 1:50, and 1:60 g/mL, the extraction temperature was set to 30, 40, 50, 60, 70, and 80 °C, and the extraction power was set to 50, 60, 70, 80, 90, and 100 W.

#### 3.2.4. Optimization of LAH Extraction Process Using Response Surface Methodology

Based on the results of the one-factor experiment, four factors, including ethanol concentration (A), material-to-liquid ratio (B), extraction temperature (C), and extraction power (D), were selected as variables. Design-Expert 8.0.6 software was used to conduct the three-level response surface experiment of the four factors, with −1, 0, and 1 representing variable levels, to investigate the effects of different level combinations of the four factors on the extraction rate of LAH. The real values and coding levels of independent variables are shown in Table 3.

#### 3.2.5. Verification Test

Four samples of LAH were extracted according to the modified extraction conditions, and accurately weighed, at 1.0 g each; then the extraction rate was calculated.

### 3.3. Quasi-Targeted Metabolomics Analysis

The samples of LAH that were extracted via the optimized process were analyzed using quasi-targeted metabolomics (Beijing Novogene Co., Ltd., Beijing, China). The LAH extract was dissolved in 80% methanol aqueous solution, the supernatant was taken after centrifugation with shaking and freeze-dried into dry powder, and then the corresponding 10% aqueous methanol solution was added to the volume of sample taken, dissolved, and analyzed via LC-MS. An equal volume of each sample was mixed as the quality control (QC) sample, and 53% aqueous methanol solution replaced the experimental sample (the same treatment procedure as that of the experimental sample) as the blank control. LC-MS was performed on an Xselect HSS T3 column (2.5 μm, 2.1 × 150 mm), with the mobile phases of 0.1% formic acid-water (A) and 0.1% formic acid-acetonitrile (B) at the flow rate of 0.4 mL/min and the column temperature of 50 °C. The chromatographic gradient elution procedure is shown in Table 4, and the mass spectrometry conditions are shown in Table 5. Finally, the samples were detected using the multiple reaction monitoring (MRM) mode based on the highly sensitive SCIEX QTRAP^®^ 6500+ mass spectrometry platform in Novogene Co., Ltd. (Beijing, China). The compounds were quantified according to Q3 (daughter ion) and characterized by Q1 (parent ion), Q3 (daughter ion), RT (retention time), DP (de-clustering voltage), and CE (collision energy).

### 3.4. Cell Culture

The RAW264.7 macrophages were cultured in DMEM, supplemented with 10% FBS (HyClone, Logan, UT, USA), at 37 °C in a fully humidified incubator containing 5% CO_2_. When the cells grew into a dense monolayer, cell passaging was performed by the conventional method.

### 3.5. Cell Viability Assay

The cell viability was determined via the CCK-8 assay according to the manufacturer’s instructions and a previous procedure with minor modifications [39]. The RAW264.7 cells were seeded in 96-well plates (Eppendorf, Hamburg, Germany) at a density of 6 × 10^4^/mL in a culture medium overnight. Then, the cells were treated with DMEM containing 0 to 200 µg/mL of LAH. After 24 h of incubation, the original medium was discarded, and 100 µL of 10% CCK-8 solution was added to each well. The absorbance values were measured at 450 nm 4 h after CCK-8 addition using a spectrophotometer (Epoch Microplate Spectrophotometer, BioTek Instruments, Inc., Winooski, VT, USA). The cell viability was expressed as the percentage viability according to the following formula:Cell Viability (%) = [(absorbance of treatment − absorbance of blank)/(absorbance of control − absorbance of blank)] × 100%

### 3.6. ELISA

The RAW264.7 cells were seeded in 6-well plates (Eppendorf, Hamburg, Germany) at a density of 10^6^/mL in a culture medium overnight. Then, the cells were treated with DMEM containing different concentrations of LAH (0, 10, 30, and 50 µg/mL) for 12 h. The positive control group and LAH-treated groups were then exposed to LPS (10 µg/mL) for 12 h. The medium of each group was collected and centrifuged at 4 °C with 1000 g for 10 min. The obtained supernatant was analyzed using ELISA kits according to the manufacturer’s instructions. The absorbance values were measured at 450 nm using a spectrophotometer (Epoch Microplate Spectrophotometer, BioTek Instruments, Inc., Winooski, VT, USA), and the concentrations of IL-4, IL-10, IFN-γ, IL-6, and TNF-α in the samples were calculated according to the standard curve.

### 3.7. Network Pharmacology Analysis

#### 3.7.1. Screening for Active Ingredients of LAH

All the chemical constituents of LAH extract identified via quasi-targeted metabolomics analysis were screened according to oral bioavailability (OB), drug-likeness (DL), and relative contents, and the screening thresholds of each component were set as OB ≥ 30%, DL ≥ 0.18, and relative contents > 1 × 10^6^, respectively. The SMILE structures of the screened ingredients were obtained through the PubChem database (https://pubchem.ncbi.nlm.nih.gov/, accessed on 17 May 2022) [40], and their targets were searched through the SwissTarget Prediction database (http://www.swisstargetprediction.ch/, accessed on 19 May 2022) [41] for subsequent analysis.

#### 3.7.2. Construction of Common Targets between Active Ingredients of LAH and Diseases

The keyword “anti-inflammatory” was used to search disease-related genes on the GeneCards database (https://www.genecards.org/, accessed on 19 May 2022) [42]. The targets of the bioactive ingredients of LAH were mapped to the target genes related to “anti-inflammatory” to obtain the common target genes through the online tool “jvenn” (http://jvenn.toulouse.inra.fr/app/example.html, accessed on 19 May 2022).

#### 3.7.3. Construction of Protein–Protein Interaction Network

Based on the overlap of predicted targets of LAH and anti-inflammatory related targets, a protein–protein interaction (PPI) network was constructed on the STRING database (https://string-db.org/, accessed on 24 May 2022) [43] to further elucidate the potential anti-inflammatory mechanism of LAH. With the setting conditions of “Homo sapiens”, “Hide disconnected nodes in the network”, and “Minimum required interaction score = 0.4”, the protein interaction information including the node degree value was obtained. Cytoscape 3.7.2 software [44] was used to visualize the PPI network, and the “network Analysis” plug-in was used to analyze the topological properties of each node and select the hub gene according to the obtained node degree values.

#### 3.7.4. Analyses of GO and KEGG Pathway Enrichment

The DAVID database (https://david.ncifcrf.gov/home.jsp, accessed on 27 May 2022) [45] was used to analyze Gene Ontology (GO) enrichment in biological function/process (BP), cellular component (CC), and molecular function (MF), and Kyoto Encyclopedia of Genes and Genomes (KEGG) pathway enrichment with the “Homo sapiens” setting (adjusted to *p* < 0.05). The visualized bubble charts and histograms were formed using the R-based online graphing tools “Bioinformatics” (https://www.bioinformatics.com.cn, accessed on 27 May 2022) and “ChiPlot” (https://www.chiplot.online/, accessed on 27 May 2022).

### 3.8. Western Blotting Analysis

The culture medium of each treatment group was discarded, and RIPA lysis buffer containing PMSF was added. The cells were gently scraped, collected into centrifuge tubes, and centrifuged at 4 °C with 12,000 rpm for 30 min. The concentrations of total protein of the collected supernatants were measured using the BCA protein assay kit. Equal amounts of protein lysates were separated by SDS-PAGE (10% gel), and the gel was transferred to NC membranes, washed with TBST buffer, blocked with QuickBlock™ blocking buffer for 15 min, and incubated with the corresponding antibody solution (1:1000) for 12–16 h at 4 °C. After washing, the NC membranes were incubated with the secondary antibody at 37 °C for 1 h. Chemiluminescence positive signals were detected using the ECL Western blotting detection reagent (Thermo Scientific, Waltham, MA, USA). Protein band images were scanned, the integrated absorbance (IA) of the target bands was analyzed using ImageJ software (version No. 1.53e), and the relative level of the target protein was normalized to β-actin (target protein IA/β-actin IA).

### 3.9. Statistical Analysis

Statistical analyses were performed via the one-way ANOVA in SPSS 26.0 for Windows (SPSS Inc., Chicago, IL, USA). The experimental data were expressed as the mean ± standard deviation (SD) of three independent experiments. *p* < 0.05 was considered statistically significant.

## 4. Discussion

The chemical composition of plants includes primary metabolites (e.g., proteins, fats, and sugars) common to most organisms, as well as secondary metabolites produced through a variety of metabolic pathways within plant cells [46]. The secondary metabolites that are isolated from plant extracts are usually referred to as natural products and include alkaloids, flavonoids, saponins, terpenoids, sterols, etc. [47]. Compared with traditional extraction methods such as Soxhlet extraction, maceration, percolation, and decoction, unconventional extraction techniques such as supercritical fluid extraction (SFE), microwave-assisted extraction (MAE), and ultrasound-assisted extraction (UAE) offer shorter extraction times and higher extraction efficiencies, and at the same time, are more conducive to the maintenance of the stability of the natural products and easier for application in industrial production [48,49]. Among them, UAE induces cavitation by destroying the plant cell wall with high-frequency pulses, which makes it easier for the solvent to penetrate into the plant cells, thus increasing the rate of delivery of bioactive components to the solvent [50,51]. Thus, UAE has the characteristics of simple operation, low cost, ease of commercial integration, and optimization [52]. In this study, UAE was selected for the extraction of LAH, and the process parameters of UAE were optimized using the response surface method. The optimized extraction process conditions were an ethanol concentration of 72%, a material-to-solvent ratio of 1:37 g/mL, an extraction temperature of 73 °C, and an extraction power of 70 W. The extraction rate of total LAH flavonoids extracted by this process condition was 0.378%, which was not much different from the theoretical prediction value. In the process of ultrasound-assisted extraction of LAH, the extraction method, the number of extraction times, and the extraction solvent selection all affect the extraction rate of LAH total flavonoids. Therefore, the consideration of a single-factor experiment in this study was not comprehensive enough, which was the reason for some limitations of this study, and this could be optimized in future studies.

LAH contains thousands of chemical components, which are difficult to be accurately analyzed at once by conventional techniques. Therefore, a quasi-targeted metabolomics analysis was used in this study to characterize and quantify the metabolites in LAH. As a novel metabolomics detection technique, quasi-targeted metabolomics has been widely used in the study of natural products. Cao et al. adopted quasi-targeted metabolomics technology in the study of ethanol extract from passion fruit peel, and identified 91 kinds of flavonoids, 84 kinds of phenolic compounds, and 3 kinds of anthocyanins in the extract [53]. Wang et al. provided a reference for related studies by establishing a quasi-targeted metabolomics approach to accurately identify chemical compounds in licorice, and the qualitative results identified a total of 1435 compounds including compounds in licorice and possible probiotic metabolites [54]. The results of the quasi-targeted metabolomics analysis of LAH in this study showed that a total of 1666 compounds were identified in LAH, and their relative contents were used as screening parameters for subsequent studies. Although the quasi-targeted metabolomics analysis provided some information on the components contained in LAH, more in-depth studies are still needed to identify the specific components and their percentages that exert pharmacological activities in LAH. In future studies, the isolation and purification of the extracts and the application of chromatographic and spectroscopic techniques will help to explore the specific information of the pharmacologically active components in LAH extracts in greater depth.

Macrophages exist widely in lymphoid tissue and are the organisms’ first line of immune defense against infection [55]. Macrophages play an important role in regulating the occurrence, development, and extinction of inflammatory responses due to their diversity and heterogeneity in response to changes in the microenvironment [56,57]. Toll-like receptors (TLRs), which are recognition receptors that recognize pathogen-associated molecules, can interact with pathogen-derived components (e.g., LPS), resulting in the up-regulation of macrophage function and further mediation of the inflammatory responses [58,59]. Therefore, macrophages are important targets for inflammatory therapeutic drug research. In this study, the cytotoxicity of LAH on macrophages was detected via the CCK-8 assay, and the results showed that LAH at concentrations below 100 μg/mL has no cytotoxicity. Therefore, 10, 30, and 50 μg/mL of LAH were selected for subsequent experiments. It is noteworthy that 10–50 µg/mL of LAH significantly promoted cell proliferative activity. Studies have shown that bioactive compounds in natural products have a promoting effect on the expression of growth factors associated with cell proliferation, e.g., the ethanol extract of *Hericium erinaceus* enhanced the synthesis of neurotrophic factor (NGF) through the JNK pathway, which promoted the proliferation of hippocampal neural stem cells [60]. It is hypothesized that the promoting effect of LAH on cell proliferation may be related to its promotion of growth factor expression. Fu et al. investigated the effects of Cibotium barometz polysaccharides (CBPS) on the viability and cell cycle transition of primary chondrocytes in rats, and found that CBPS had an enhancing effect on chondrocyte proliferation, which may be related to its promotion of G1/S cell cycle transition [61]. Therefore, the regulatory effect of LAH on the cell cycle may also be involved in its promotion of cell proliferation. Furthermore, the natural product Loureirin B (LB) has been shown to promote the proliferation of Ins-1 cells while inhibiting their apoptosis [62]. It is inferred that the role of LAH in promoting cell proliferation is also related to its regulation of apoptosis. Astragalus polysaccharide (APS) has been shown to reduce the levels of ROS, MDA, and NO in diabetic cardiomyopathy (DCM) cell models, improve the activity of antioxidant enzymes, and promote the proliferation of DCM cells, thus playing a protective role in DCM cells, which may be related to the activation of the NGR1/ErbB signaling pathway by APS [63]. LAH has been reported to have favorable antioxidant activity in previous studies [64], and thus its modulation of oxidative metabolism may also be involved in its promotion of cell proliferation.

Macrophages can be polarized into a pro-inflammatory M1 phenotype and an anti-inflammatory M2 phenotype, and these two phenotypes can be transformed into each other in response to changes in the microenvironment [65]. Previous studies have demonstrated that LPS, as an important inflammatory trigger, can be recognized by the TLR4 receptor on the surface of macrophages when it invades the organism, resulting in the activation of macrophages toward the M1 phenotype and the secretion of a large number of pro-inflammatory cytokines, such as IL-6, TNF-α, and IFN-γ, which leads to the further activation of other inflammatory cells and the expansion of inflammatory responses [56,66]. Thus, one of the key factors in the development of inflammatory diseases is the large amount of pro-inflammatory cytokines secreted by the M1 phenotype of macrophages [67]. In contrast, the M2 phenotype of macrophages is characterized by the secretion of anti-inflammatory cytokines such as IL-10 and Arg-1, which contribute to tissue repair and defense against excessive inflammation [68]. Zhao et al. investigated the pharmacological effects of the traditional Chinese medicine HuoXueTongFu Formula (HXTF), and found that macrophages produced large amounts of pro-inflammatory cytokines after LPS stimulation, whereas the levels of anti-inflammatory cytokines, such as IL-4 and IL-10, were significantly decreased. The intervention of HXTF significantly enhanced the expression of anti-inflammatory factors and decreased the expression of pro-inflammatory factors [69]. Consistent with previous findings, our study showed that macrophages exposed to LPS significantly secreted the pro-inflammatory cytokines IL-6, TNF-α, and IFN-γ, while LAH treatment significantly reversed this effect of LPS. Among them, the release of pro-inflammatory cytokines was significantly reduced in the medium-dose and high-dose groups of LAH compared with the LPS group (*p* < 0.05). In addition, the concentrations of the anti-inflammatory cytokines IL-4 and IL-10 were significantly decreased in the LPS-treated group compared with the blank control group (*p* < 0.01). In contrast, the concentrations of anti-inflammatory cytokines IL-4 and IL-10 in the supernatants of the cell culture medium increased significantly (*p* < 0.05) after treatment with LAH. Therefore, LAH may exert its anti-inflammatory effects by inhibiting the secretion of pro-inflammatory cytokines and promoting the secretion of anti-inflammatory cytokines.

Network pharmacology is an emerging discipline proposed by Hopkins in 2007 [70], which effectively reveals the relationship between drugs and diseases through the comprehensive use of multidisciplinary technologies and methods, and its application plays an important role in analyzing the molecular mechanism of natural products acting on diseases. In this study, we first used the high-throughput Chinese medicine database to query the relevant information of 1666 substances identified by the quasi-targeted metabolomics analysis, and screened the active ingredients in LAH according to the OB, DL values, and relative content of the substances. Oral bioavailability (OB) represents the orally administered percentage of a drug which reaches the site of action without changes. Good oral bioavailability is one of the key pharmacokinetic parameters for screening new drug candidates since the oral route is the most convenient and predominant mode of administration [71]. Drug-likeness (DL) is the index which represents the balance between pharmacodynamics affected by molecular properties and the pharmacokinetics of molecules, which is used to estimate the degree of “drug-like” of the candidate components. The average DL index in the Drugbank database is 0.18, which is regarded as the selection standard for the candidate components in the traditional Chinese medicine [72,73]. In this study, the relevant information of 1666 substances identified by the quasi-targeted metabolomics analysis was queried using the high-throughput TCM database, and the active ingredients in LAH were screened according to the OB, DL, and relative content values of the substances. At the same time, it was considered that some substances did not meet the screening criteria but have been reported to have anti-inflammatory activity. Therefore, a total of 96 active ingredients of LAH were screened for follow-up studies based on the above conditions. Subsequently, a total of 255 anti-inflammatory targets of LAH were obtained by searching and analyzing the active ingredient targets in several network-pharmacology-related databases. Through the GO enrichment analysis and KEGG pathway analysis of these targets, the relevant information of biological processes, cell components, molecular functions, and signaling pathways involved in the anti-inflammatory effect of LAH was obtained. In addition, through the construction and analysis of the PPI network, hub genes including AKT1, MAPK3, RELA, PTGS2, and JUN were screened in order to further validate and analyze the anti-inflammatory mechanism of LAH.

AKT (AKT1, AKT2, and AKT3) is a serine/threonine protein kinase, also known as protein kinase B (PKB), which can be activated by a variety of extracellular stimuli to play an important role in cellular responses including cell survival, metabolism, differentiation, and proliferation [74,75]. It has been reported that AKT1 and AKT2 are involved in inflammatory responses in various tissues and diseases [75]. Among them, Akt1 is involved in pro-inflammatory signaling in macrophages and regulates macrophage innate immune function by mediating the generation of mitochondrial H_2_O_2_; its activation is associated with inflammation, oxidative stress, and the accumulation of oxidized lipids [74,76,77]. AKT regulates the release of inflammatory factors and has been shown to have an important role in a variety of disease models, such as hepatic ischemia-reperfusion injury, acute lung injury, and hypoxic-ischemic encephalopathy [78,79]. Meng et al. investigated the anti-neuroinflammatory effects and mechanisms of Evodiamine (EV) by establishing a model of LPS-induced inflammation of BV-1 cells, and showed that EV alleviated the neuroinflammatory response by affecting LPS-induced AKT phosphorylation [80]. In addition, in vivo and in vitro experiments have shown that Betulin, a natural product, plays an anti-inflammatory role by regulating the AKT signaling pathway to enhance the expression of anti-inflammatory genes in the nucleus and inhibit the expression and secretion of pro-inflammatory cytokines [81]. Consistent with the above studies, our results showed that LAH significantly inhibited the protein expression of AKT in the LPS-induced macrophage inflammation model in all dose groups (*p* < 0.01). Thus, the alleviating effect of LAH on macrophage inflammatory response is related to its blocking of the AKT signaling pathway.

RELA (also known as p65 NF-κB) is one of the subunits of the nuclear transcription factor NF-κB complex, which plays an important regulatory role in inflammatory processes [82]. Activated RELA translocates to the nucleus and acts as a transcription factor that initiates inflammatory genes to promote the production of various cytokines and chemokines in inflammatory states [83]. Therefore, the RELA signaling pathway has been identified as a major contributor to the inflammatory response and may serve as a key therapeutic target for inflammatory diseases [84]. Activated AKT affects the activity of nuclear transcription factor inhibitor kinase through direct or indirect pathways, thereby regulating RELA activation, nuclear translocation, and the RELA-dependent transcription of other genes. In addition, the direct promotion of nuclear transcription factor inhibitor phosphorylation is also one of the pathways by which AKT activates RELA [85,86,87]. It has been reported that under inflammatory conditions, the over-activation of AKT1 leads to the massive production of pro-inflammatory cytokines, which aggravates the inflammatory response [76,88]. The LPS-induced inflammatory response in macrophages is also enhanced by the up-regulation of the NF-κB signaling pathway by activated AKT [89]. Therefore, the inhibition of AKT and RELA signaling is critical for the improvement of pathophysiological conditions including cancer, neuroinflammation, and diabetes [90,91]. In a study of the pharmacological effects and mechanisms of the mitochondria-targeted antioxidant N, N’-Bis(salicylideneamino)ethane-manganese (II)-8 (EUK-8), Jayachandra et al. found that it is not only a powerful antioxidant but also a multi-targeted inflammatory-pathway-inhibitory molecule. In addition to inhibiting pro-inflammatory cytokines and chemokines in edematous tissues, EUK-8 also could control the progression of inflammation by down-regulating the protein expression levels of AKT and RELA [92]. Similarly, the results of our experiments showed that LAH significantly inhibited the protein expression level of RELA (*p* < 0.01) with a certain dose-dependent trend. Therefore, the anti-inflammatory role played by LAH in the development of LPS-induced macrophage inflammation may be related to its regulation of AKT and RELA.

PTGS2 (also known as COX-2) is an inducible enzyme not expressed in normal cells. It acts as a major pro-inflammatory protein and is essential for the synthesis of the prostaglandin associated with pathological processes, especially acute and chronic inflammation [84,93]. The high expression of PTGS2 can be detected when immune cells are stimulated by pro-inflammatory cytokines, LPS, and carcinogens. Studies have shown that AKT1 plays a key role as a regulator in LPS-induced PTGS2 expression [94]. AKT1 promotes the translocation of the transcription factor NF-κB from the cytoplasm to the nucleus through the activation of the NF-κB pathway, thereby regulating the expression of inflammatory mediators such as PTGS2, iNOS, IL-1β, IL-6, and TNF-α [80]. In view of the important roles of AKT, RELA, and PTGS2 in the inflammatory response process, Fang et al. investigated their roles in the alleviation of LPS-stimulated macrophage inflammation by Plantanone C (PC). The results showed that PC alleviated LPS-induced macrophage inflammation by blocking the AKT signaling pathway and attenuating the expression of LPS-activated RELA and PTGS2, thereby inhibiting the massive secretion of inflammatory mediators and proinflammatory cytokines [95]. Consistent with the above studies, our study showed that RAW264.7 cells secreted a large number of pro-inflammatory cytokines after LPS treatment, and the protein expression levels of AKT1, RELA, and PTGS2 were significantly increased (*p* < 0.05). However, each dose of LAH could inhibit their protein expression levels to varying degrees (*p* < 0.05), and inhibit the secretion of pro-inflammatory cytokines at the same time. In summary, LAH may regulate the expression of pro-inflammatory mediators and the secretion of pro-inflammatory cytokines through the AKT1/RELA/PTGS2 signaling pathway, thereby alleviating LPS-induced inflammatory injury in macrophages.

Accumulated evidence suggests that the up-regulation of the activity of non-receptor tyrosine kinases, including AKT, SYK, SRC, JAK, PI3K, and MAPKs, is also involved in mediating the overproduction of inflammatory mediators [96]. Among them, MAPKs consist of three isoforms, JNK, MAPK3 (ERK1), and p38 kinase, which are members of the highly conserved serine/threonine protein kinase family [97]. As known up-regulators of NF-κB, MAPKs play a critical control role in processes associated with cellular transduction including inflammation, cancer metastasis, and cell proliferation [98]. Phosphorylated MAPKs bind and activate target kinases, and then translocate to the nucleus to further activate the transcription of pro-inflammatory genes. It has been reported that MAPKs activate intracellular signal transduction in the NF-κB pathway, thereby inducing the expression of inflammatory mediators such as IL-6 and TNF-α [99]. In addition, MAPKs and NF-κB signaling pathways have also been demonstrated to be involved in the TNF-α stimulation process of several cytokines that contribute to the various inflammatory pathogenesis [100]. It has been shown that signaling cascades including MAPKs and AKT are activated upon LPS binding to the TLR4 receptor, which directly promotes DNA-binding activity in macrophages by phosphorylating transcription factors and activating downstream signaling pathways such as NF-κB and AP-1, thereby up-regulating the mRNA expression levels of inflammatory mediators such as TNF-α and iNOS [101,102,103]. The AP-1 transcription factor, a downstream target of MAPKs, is a homologous or heterodimeric protein complex composed of the FOS (c-Fos, FosB, Fra-1, and Fra-2) and JUN (c-Jun, JunB, and JunD) subfamilies [104,105]. As regulators of the AP-1 transcription factor, activated MAPKs stimulate the activation of the transcription factor JUN and nuclear translocation, a process that has also been shown to be associated with pro-inflammatory gene expression [106]. The study of MAPK/AP-1 signaling in airway epithelial cells has revealed that the activation of TLR is associated with this signaling pathway and may contribute to the production of inflammatory mediators and host defense proteins [107]. Therefore, MAPKs regulate the production of pro-inflammatory mediators and cytokines, and influence cellular responses to environmental stresses by activating AP-1 [108,109]. Studies have shown that AP-1 plays a critical mediated role as the key transcription factor in the transcriptional activation of multiple inflammatory genes [110]. Specifically, AP-1 is located in the cytoplasm under normal physiological conditions and is induced to translocate to the nucleus upon the activation of MAPKs by inflammatory stimuli such as LPS. The inflammatory signaling cascade is thus initiated, which in turn affects the expression of inflammatory mediators PTGS2 and iNOS, and pro-inflammatory cytokines IL-1β, IL-6, and TNF-α [111]. In addition, it has been reported that PTGS2 expression can be post-transcriptionally regulated by MAPKs through the activation of AP-1 complexes or mRNA stabilization [112].

Mitra et al. demonstrated through in vivo and in vitro experiments that Korean Red Ginseng water extract had a significant inhibitory effect on cadmium-induced pro-inflammatory cytokine production by inactivating the MAPKs/AP-1 signaling pathway [113]. Sakai et al. reported that astaxanthin could block the DSS-induced translocation of RELA and JUN into the nucleus of mucosal epithelial cells, while inhibiting the activation of MAPKs in mucosal cells, thereby significantly reducing the mRNA expression levels of the pro-inflammatory cytokines IL-1β, IL-6, and TNF-α, and exerting a mitigating effect on inflammatory bowel disease [111]. Moreover, Abekura et al. showed that the inhibition of the MAPK, NF-κB, and AP-1 signaling pathways significantly down-regulated the expression levels of pro-inflammatory cytokines and inflammatory mediators (PTGS2, NO), and thus suppressed LPS-induced inflammatory responses in murine macrophages [114]. Our findings are consistent with those reported above that the protein expression levels of MAPK3 and JUN were significantly elevated in LPS-stimulated macrophages compared with the blank control group (*p* < 0.01), whereas LAH significantly reversed this trend (*p* < 0.01). At the same time, the secretion levels of pro-inflammatory cytokines and the protein expression levels of the pro-inflammatory mediator PTGS2 in all three LAH dosage groups were significantly down-regulated. Thus, our data strongly suggested that LAH may have a role in alleviating macrophage inflammation by inhibiting the activation of the LPS-induced MAPK3/JUN signaling pathway, while down-regulating the levels of pro-inflammatory cytokines and inflammatory mediators. 

The mechanism of LAH exerting anti-inflammatory effects in on LPS-induced inflammatory macrophages was shown in Figure 6. In addition, there still exist some limitations in this study. For example, AKT1, PTGS2, RELA, JUN, and MAPK3 were analyzed using network pharmacology analysis as potential targets of LAH, and our data also demonstrated that LAH could regulate their expression. However, these findings could not demonstrate a direct interaction between LAH and AKT1, PTGS2, RELA, JUN, and MAPK3. The use of selective inhibitors and fluorescently labeled probes will help to validate the above questions. Furthermore, the anti-inflammatory effect of LAH was tested in vitro by establishing a macrophage inflammation model in this study, and further experiments are still required to prove whether LAH has an anti-inflammatory effect in vivo.

## Figures and Tables

**Figure 1 ijms-24-16272-f001:**
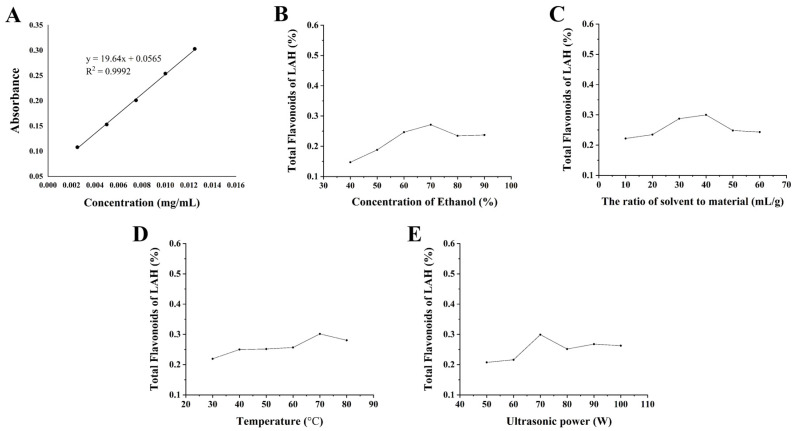
(**A**) Standard curve and equation of quercetin. (**B**) The effect of the ethanol concentration on the LAH extraction amount. (**C**) The effect of the material-to-solvent ratio on the LAH extraction amount. (**D**) The effect of the extraction temperature on the LAH extraction amount. (**E**) The effect of the extraction power on the LAH extraction amount.

**Figure 2 ijms-24-16272-f002:**
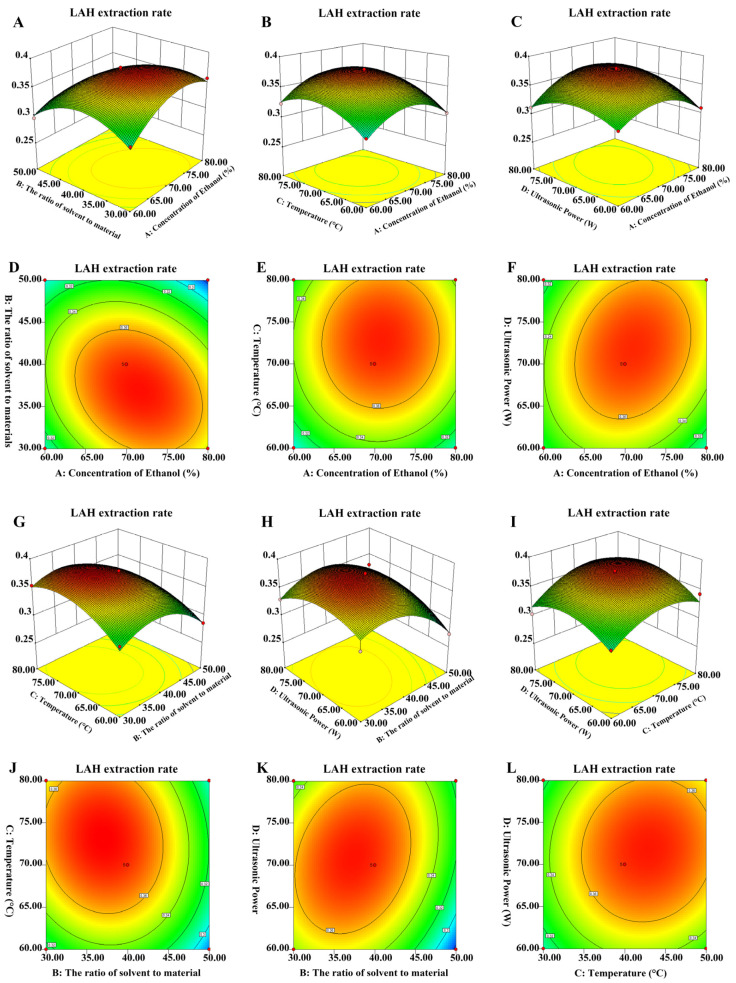
Response surface and the contour diagram of the interaction of different factors to the LAH extraction rate. (**A**,**D**) Response surface and the contour plot of the interaction of ethanol concentration and material-to-solvent ratio to the LAH extraction rate. (**B**,**E**) Response surface and the contour plot of the interaction of extraction temperature and ethanol concentration to the LAH extraction rate. (**C**,**F**) Response surface and the contour plot of the interaction of extraction power and ethanol concentration to the LAH extraction rate. (**G**,**J**) Response surface and the contour plot of the interaction of extraction temperature and material-to-solvent ratio to the LAH extraction rate. (**H**,**K**) Response surface and the contour plot of the interaction of extraction power and material-to-solvent ratio to the LAH extraction rate. (**I**,**L**) Response surface and the contour plot of the interaction of extraction power and extraction temperature to the LAH extraction rate.

**Figure 3 ijms-24-16272-f003:**
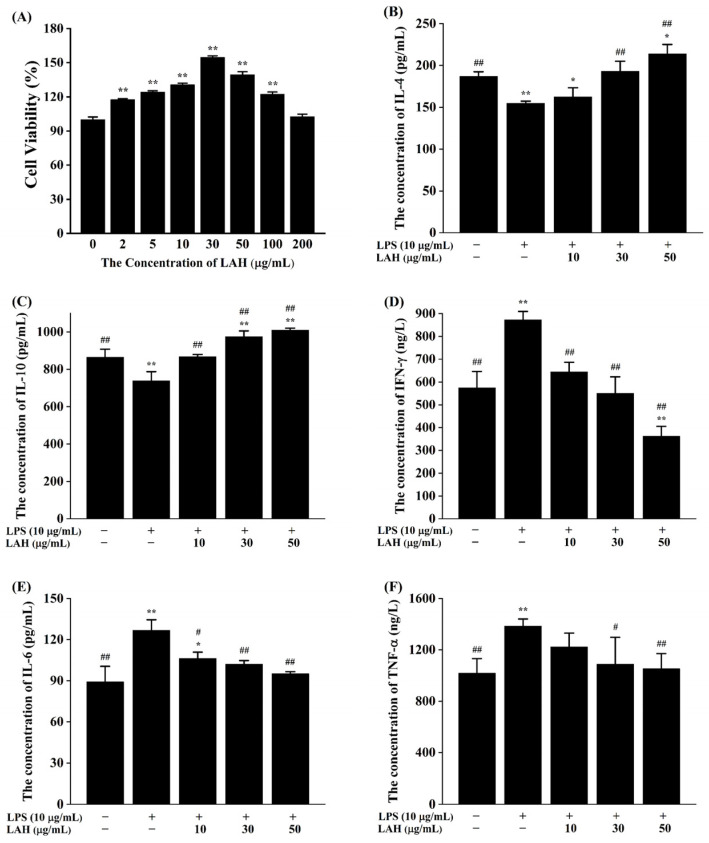
(**A**) The effects of LAH on the cell viability. (**B**–**F**) The effects of LAH on cellular secretion of IL-4, IL-10, IFN-γ, IL-6, and TNF-α. The results are expressed as the mean ± standard deviation of three independent experiments. * *p* < 0.05 and ** *p* < 0.01 compared with the blank control group, # *p* < 0.05 and ## *p* < 0.01 compared with the LPS group were considered statistically significant differences.

**Figure 4 ijms-24-16272-f004:**
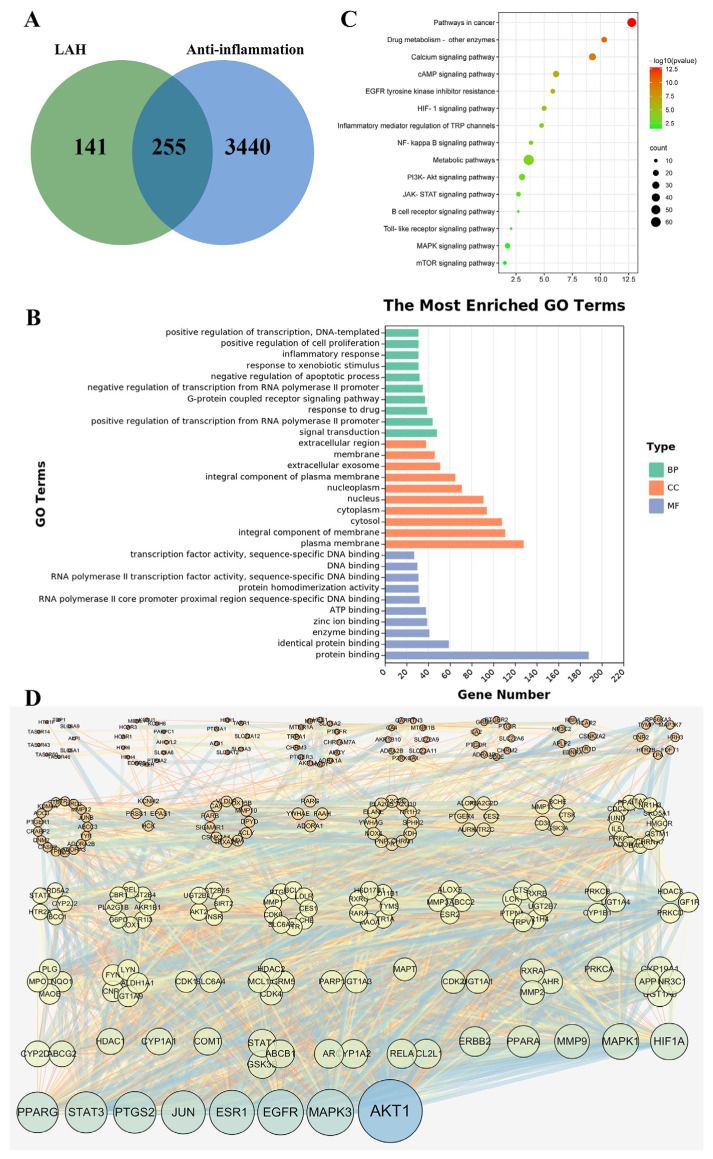
Network pharmacology analysis of the anti-inflammatory effects of LAH. (**A**) Venn diagram of the interactions between all candidate targets of LAH and anti-inflammatory targets. (**B**) Histogram of GO enrichment analysis of the anti-inflammatory targets of LAH. (**C**) Bubble chart of KEGG pathway enrichment analysis of the anti-inflammatory targets of LAH. (**D**) PPI network of the anti-inflammatory targets of LAH, in which targets are represented by nodes, interactions between targets are represented by edges, colors and sizes of targets are represented according to the target’s degree from dark to light and from large to small, respectively, and the combined score of targets is indicated by the thickness of the edges.

**Figure 5 ijms-24-16272-f005:**
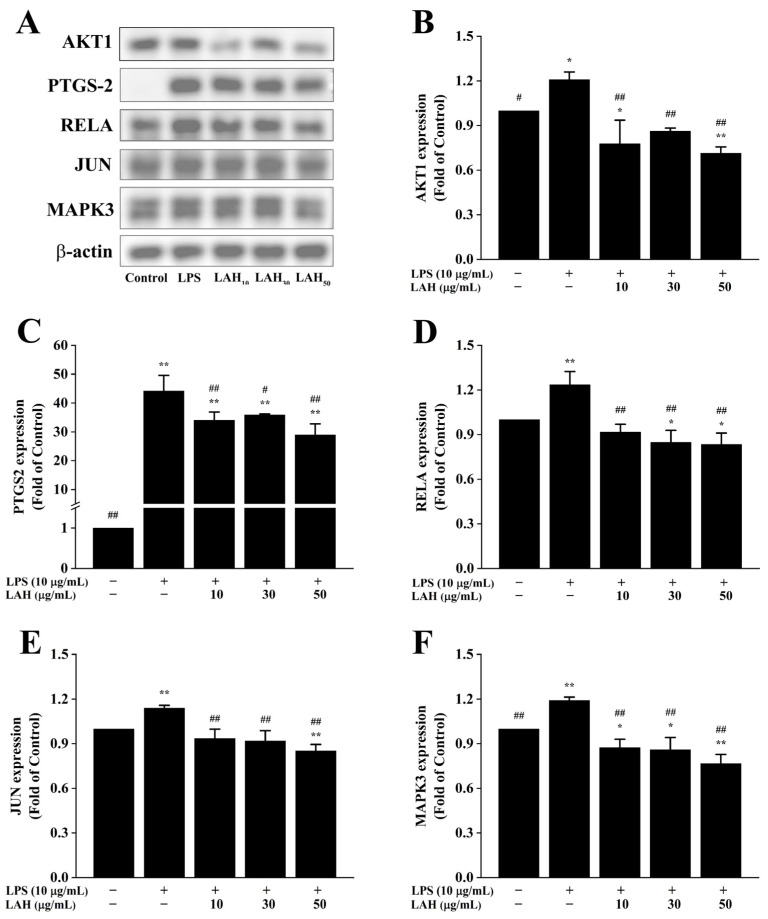
Representative protein bands and the histograms of band intensity analysis of hub genes in LPS-induced RAW264.7 cells. (**A**) Representative protein bands of AKT1, PTGS2, RELA, JUN, and MAPK3. (**B**) The histograms of band intensity analysis of AKT1. (**C**) The histograms of band intensity analysis of PTGS2. (**D**) The histograms of band intensity analysis of RELA. (**E**) The histograms of band intensity analysis of JUN. (**F**) The histograms of band intensity analysis of MAPK3. The results are expressed as the mean ± standard deviation of three independent experiments. * *p* < 0.05 and ** *p* < 0.01 compared with the blank control group, # *p* < 0.05 and ## *p* < 0.01 compared with the LPS group were considered statistically significant differences.

**Figure 6 ijms-24-16272-f006:**
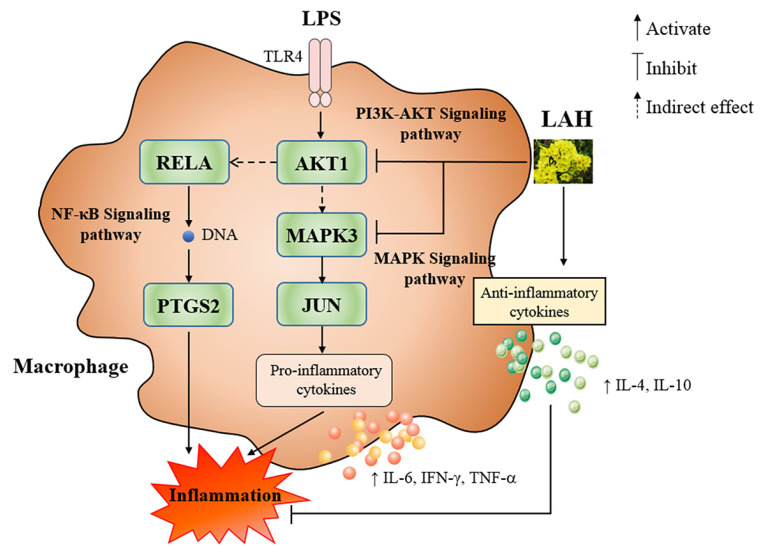
The mechanism of LAH exerting anti-inflammatory effects on LPS-induced inflammatory macrophages.

**Table 1 ijms-24-16272-t001:** Variance analysis of quadratics simulation of response surface.

Variance Sources	Sum of Squares	Degree of Freedom	Mean Square	F-Value	*p*-Value
Model	0.024	14	1.748 × 10^−3^	19.21	<0.0001
A-Ethanol concentration	1.880 × 10^−4^	1	1.880 × 10^−4^	2.07	0.1725
B-Material-to-solvent ratio	4.808 × 10^−3^	1	4.808 × 10^−3^	52.85	<0.0001
C-Extraction temperature	2.211 × 10^−3^	1	2.211 × 10^−3^	24.31	0.0002
D-Extraction power	5.658 × 10^−4^	1	5.658 × 10^−4^	6.22	0.0258
AB	1.092 × 10^−3^	1	1.092 × 10^−3^	12.01	0.0038
AC	1.482 × 10^−5^	1	1.482 × 10^−5^	0.16	0.6926
AD	5.040 × 10^−4^	1	5.040 × 10^−4^	5.54	0.0337
BC	1.134 × 10^−4^	1	1.134 × 10^−4^	1.25	0.2830
BD	6.452 × 10^−4^	1	6.452 × 10^−4^	7.09	0.0186
CD	5.852 × 10^−5^	1	5.852 × 10^−5^	0.64	0.4359
A^2^	6.135 × 10^−3^	1	6.135 × 10^−3^	67.43	<0.0001
B^2^	8.091 × 10^−3^	1	8.091 × 10^−3^	88.93	<0.0001
C^2^	4.203 × 10^−3^	1	4.203 × 10^−3^	46.20	<0.0001
D^2^	3.377 × 10^−3^	1	3.377 × 10^−3^	37.12	<0.0001
Residual	1.274 × 10^−3^	14	9.098 × 10^−5^		
Lack of fit	1.148 × 10^−3^	10	1.148 × 10^−4^	3.65	0.1118
Pure error	1.258 × 10^−4^	4	3.144 × 10^−5^		
Cor total	0.026	28			

**Table 2 ijms-24-16272-t002:** The validation test results of the optimal process conditions for LAH extraction.

Sample Weight (g)	Content of Total Flavonoids of LAH (%)	Average Value (%)	RSD (%)
1.004	0.3738	0.378	3.3
1.005	0.3755		
1.006	0.3636		
1.004	0.3976		

**Table 3 ijms-24-16272-t003:** The experimental factors and levels.

Levels	Factors			
	(A) Ethanol Concentration (%)	(B) Material-to-Liquid Ratio (g/mL)	(C) Extraction Temperature (°C)	(D) Extraction Power (W)
1	80	1:50	80	80
0	70	1:40	70	70
−1	60	1:30	60	60

**Table 4 ijms-24-16272-t004:** The chromatographic gradient elution procedure.

Time (min)	A (%)	B (%)
0	98	2
2	98	2
15	0	100
17	0	100
17.1	98	2
20	98	2

**Table 5 ijms-24-16272-t005:** The mass spectrometry conditions.

	ESI (+)	ESI (−)
Curtain Gas (psi)	35	35
Collision Gas	Medium	Medium
Ion Spray Voltage (V)	5500	−4500
Temperature (°C)	550	550
Ion Source Gas 1	60	60
Ion Source Gas 2	60	60

## Data Availability

The data used to support the findings of this study are included within the article and the Appendix A files.

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
