# Peer review of "Study on the Alleviating Effect and Potential Mechanism of Ethanolic Extract of Limonium aureum (L.) Hill. on Lipopolysaccharide-Induced Inflammatory Responses in Macrophages"

_ijms, 2023, doi:10.3390/ijms242216272_

Round 1

Reviewer 1 Report

Comments and Suggestions for Authors

Reviewer’s Comment #:

The manuscript describes a study titled "Study on the alleviating effect and potential mechanism of Limonium aureum on LPS-induced inflammatory response in macrophages " The author's objective is to explore the anti-inflammatory properties of LAH with the goal of developing a natural anti-inflammatory drug. Overall, the manuscript is well-written and the research well-executed. The selected parameters for strategy development are both conventional and informative. However, there are a few areas that require attention to enhance the manuscript's quality and readability.

Abstract: The abstract is somewhat confusing in its current form. It should be reframed to simplify the content.

Introduction: The last paragraph of the introduction needs more clarity.

Materials and Methods: Sections 2.2.2 and 2.2.3 are somewhat complex and could benefit from simplification for better understanding.

Section 2.3: This section requires more detail.

General comments:

·      Please clarify the distinctions between conventional and unconventional extraction techniques for natural products and explain why unconventional methods, such as ultrasound-assisted extraction (UAE), are preferred.

  • Describe how UAE works and highlight its advantages in the context of natural product extraction. Have you conducted any comparisons with conventional extraction methods?
  • Provide information on the key parameters optimized during the LAH extraction process and the optimized extraction conditions.
  • Explain the criteria used to select the 96 active ingredients of LAH for further study.

Inflammatory Response Mechanisms: Elaborate on the roles of AKT, RELA, and PTGS2 in the inflammatory response and discuss how LAH influences their expression.

  • Provide additional information on the role of MAPKs and JUN in mediating the production of inflammatory mediators and detail how LAH affects their expression.

Discussion: The discussion section is weakly written and would benefit from incorporating more recent updates.

Graphical Abstract: If possible, consider providing a graphical abstract to enhance the visual representation of your research.

Conclusion: In light of some challenges encountered in reading and comprehending certain parts of the manuscript, critical corrections are needed. However, the manuscript addresses important issues, employs interesting approaches and techniques, and contributes to our understanding of LAH's role in anti-inflammatory activity and related pathways. With some minor revisions, I believe this manuscript is suitable for publication in IJMS.

Comments on the Quality of English Language

Minor corrections needed.

Author Response

Response to Reviewer 1:

Dear Reviewer,

Thank you very much for your careful and comprehensive review of our paper. Based on your comment and suggestion, we revised the relevant part in manuscript. The responses answering every question are as follows. All the lines and pages mentioned in responses are form the revised manuscript in review mode (Word version).

Comments and Suggestions.

The manuscript describes a study titled "Study on the alleviating effect and potential mechanism of Limonium aureum on LPS-induced inflammatory response in macrophages " The author's objective is to explore the anti-inflammatory properties of LAH with the goal of developing a natural anti-inflammatory drug. Overall, the manuscript is well-written and the research well-executed. The selected parameters for strategy development are both conventional and informative. However, there are a few areas that require attention to enhance the manuscript's quality and readability.

Response: Thank you for your careful and rigorous review of the article and your affirmation of our work. We have read your comments and suggestions carefully. We will carefully revise the article according to the comments and suggestions you listed. Thank you!

Comment 1. The abstract is somewhat confusing in its current form. It should be reframed to simplify the content.

Response: Thank you for your comment and suggestion. We apologize for the unclear description. We have revised the abstract in the manuscript (page 1, lines 15-41) according your advice. Thank you.

Comment 2. Introduction: The last paragraph of the introduction needs more clarity.

Response: Thank you for your comment and suggestion. We have carefully revised the last paragraph of the introduction to make it more clarity (page 3, lines 115-148). Thank you!

Comment 3. Materials and Methods: Sections 2.2.2 and 2.2.3 are somewhat complex and could benefit from simplification for better understanding.

Response: Thank you for your comment and suggestion. We apologize for any confusion our weak presentation ability may have caused you. We have revised Sections 2.2.2 and 2.2.3 in the manuscript (pages 4-5, lines 176-210). Thank you!

Comment 4. Section 2.3: This section requires more detail.

Response: Thank you for your advice. We have added more details in Sections 2.3 of the manuscript (pages 6, lines 225-248). Thank you!

Comment 5. Please clarify the distinctions between conventional and unconventional extraction techniques for natural products and explain why unconventional methods, such as ultrasound-assisted extraction (UAE), are preferred.

Response: Thank you for your comment. Conventional extraction techniques include Soxhlet extraction, decoction, solvent extraction, steam distillation extraction, percolation, etc. [1, 2]. These methods involve the diffusion of plant extracts across the glandular walls and they all require longer time to cause the rupture of plant cells [3]. Studies have shown that unconventional extraction techniques, such as UAE, can produce mechanical effects on the plant cell wall, accelerating the rupture of the cell wall and allowing the contents of the plant cell to be released into the solvent more quickly [4]. The main difference between conventional and unconventional extraction techniques is that conventional extraction techniques have high solvent consumption, high energy consumption, long time consuming and low extraction rate, while unconventional techniques have shorter extraction time, higher extraction efficiency, and at the same time are more conducive to maintaining the stability of the natural product, which is more suitable for large-scale and efficient industrial production [5]. Therefore, UAE was selected for the extraction of natural product LAH in this study.

[1] Li, F., Chen, G., Zhang, B., Fu, X. (2017). Current applications and new opportunities for the thermal and non-thermal processing technologies to generate berry product or extracts with high nutraceutical contents. Food Res Int. 100(Pt 2), 19-30. doi: 10.1016/j.foodres.2017.08.035

[2] Kumar, K., Srivastav, S., Sharanagat, V. S. (2021). Ultrasound assisted extraction (UAE) of bioactive compounds from fruit and vegetable processing by-products: A review. Ultrason. Sonochem. 70, 105325. doi: 10.1016/j.ultsonch.2020.105325

[3] Zhang, L., Liu, Z. (2008). Optimization and comparison of ultrasound/microwave assisted extraction (UMAE) and ultrasonic assisted extraction (UAE) of lycopene from tomatoes. Ultrason. Sonochem. 15(5), 731-737. doi: 10.1016/j.ultsonch.2007.12.001

[4] Chemat, S., Ahcène Lagha, Aitamar, H., Bartels, P. V., Chemat, F. (2004). Comparison of conventional and ultrasound-assisted extraction of carvone and limonene from caraway seeds. Flavour Frag. J. 19(3), 188-195. doi:10.1002/ffj.1339

[5] Fomo, G., Madzimbamuto, T. N., Ojumu, T. V. (2020). Applications of nonconventional green extraction technolo-gies in process industries: Challenges, limitations and perspectives. Sustainability. 12, 5244. doi: 10.3390/su12135244

Comment 6. Describe how UAE works and highlight its advantages in the context of natural product extraction. Have you conducted any comparisons with conventional extraction methods?

Response: Thank you for your comment. Ultrasonic wave refers to the frequency of 20-50 kHz electromagnetic wave, it is a mechanical wave that needs an energy carrier (medium) to propagate, and when it passes through the medium, it will produce two processes of expansion and compression. Ultrasonic-assisted extraction leads to cell wall destruction and particle size reduction through the strong cavitation effect, perturbation effect, high acceleration, smashing and stirring effects and other multilevel effects produced by ultrasonic radiation pressure. The mass transfer of the cell wall increases the frequency and speed of the molecular movement of the material, increases the solvent penetration, and accelerates the entry of target components into the solvent without causing changes in the structure and function of the extracts [1].

The main advantages of UAE in the extraction of natural products include: (1) Simple instrumentation. (2) Low cost of extraction. (3) Short operation time. (4) Cavitation effect enhances the polarity of the system and improves the extraction efficiency. (5) UAE allows the addition of co-extractants to further increase the polarity of the liquid phase. (6) No requirement of solvent polarity. (7) Suitable for the extraction of components that are not heat-resistant. (8) Simple extraction process that does not easily contamination of the extract.

The mechanism by which UAE extracts active compounds from natural products is based on a combination of physical mechanisms including fragmentation, erosion, sonocapillary effect, sonoporation, local shear stress, and detexturation [1]. Under the influence of these combined effects, the yield and activity of the active ingredients of natural product extracts extracted using UAE reached higher levels in a shorter period of time compared to conventional extraction methods [2]. In recent years, UAE has attracted increasing attention in studies on the extraction of natural products, including herbs, plants, vegetables, and fruits [3]. As a result, a large number of comparisons between UAE and other extraction methods have been reported. It has been shown that the extraction procedure is simpler with UAE compared to microwave-assisted extraction (MAE) and pressurized liquid extraction (PLE) with similar analytical parameters such as sensitivity, selectivity, accuracy and precision [4]. Wang et al. comparatively analyzed the properties of pectin from grapefruit peel extracted by conventional heat extraction and UAE, and the results showed that UAE extracted pectin was more thermally stable and possessed stronger antioxidant activity and lipase inhibitory activity [5]. By comparing the effects of different extraction methods on the extraction rate of phenolic compounds, total phenolic content, total flavonoid content and antioxidant capacity of the extracts from coconut husk, it was found that UAE was more efficient and the extracts extracted using UAE had higher antioxidant activity [6]. Samaram et al. evaluated different extraction methods by comparing the physicochemical properties and stability of papaya seed oil extracted by conventional extraction methods with UAE. The results showed that UAE extracted papaya seed oil had higher oxidative stability and lower unsaponifiable matters [7]. In addition, comparison of different extraction methods of freeze-dried L. camara flower showed that UAE gave the highest extraction yield and the UAE extract showed significant bacteriostatic activity against Escherichia coli, Salmonella and Staphylococcus aureus at lower concentrations [8]. In view of the advantages shown by UAE in natural product extraction, UAE was directly selected for the extraction of LAH in the present study and no comparison was made between it and conventional extraction methods. Thank you!

[1] Wen, C., Zhang, J., Zhang, H., Dzah, C. S., Zandile. M., Duan, Y., Ma, H., Luo, X. (2018). Advances in ultrasound assisted extraction of bioactive compounds from cash crops - A review. Ultrason Sonochem. 48, 538-549. doi: 10.1016/j.ultsonch.2018.07.018

[2] Soria, A.C., Villamiel, M. (2010). Effect of ultrasound on the technological properties and bioactivity of food: a review. Trends Food Sci. Technol. 21, 323-331. doi: 10.1016/j.tifs.2010.04.003

[3] Vinatoru, M., Mason, T., Calinescu, I. (2017). Ultrasonically assisted extraction (UAE) and microwave assisted extraction (MAE) of functional compounds from plant materials. TrAC Trends Anal. Chem. 97, 159-178. doi: 10.1016/j.trac.2017.09.002

[4] Chandrapala, J., Oliver, C. M., Kentish, S., Ashokkumar, M. (2013). Use of power ultrasound to improve extraction and modify phase transitions in food processing. Food Rev. Int. 29(1), 67-91. doi: 10.1080/87559129.2012.692140

[5] Wang, W., Ma, X., Jiang, P., Hu, L., Zhi, Z., Chen, J., Ding, T., Ye, X., Liu, D. (2016). Characterization of pectin from grapefruit peel: A comparison of ultrasound-assisted and conventional heating extractions. Food Hydrocolloid. 61, 730-739. doi: 10.1016/j.foodhyd.2016.06.019.

[6] Nitiwattananon, A., Thanachasai, S. (2019). Comparison of conventional and ultrasound-assisted extraction techniques for extraction of phenolic compounds from Coconut Husk. Applied Mechanics and Materials. 891, 83-89. doi: 10.4028/www.scientific.net/AMM.891.83

[7] Samaram, S., Mirhosseini, H., Tan, C. P., Ghazali, H. M. (2014). Ultrasound-assisted extraction and solvent extraction of papaya seed oil: Crystallization and thermal behavior, saturation degree, color and oxidative stability. Ind. Crop. Prod. 52, 702-708. doi: 10.1016/j.indcrop.2013.11.047

[8] Gowda, N. A. N., Gurikar, C., Anusha, M. B., Gupta, S. (2022). Ultrasound-assisted and microwave-assisted extraction, GC-MS characterization and antimicrobial potential of freeze-dried L. camara flower. J. Pure Appl. Microbio. 16(1), 526-539. doi: 10.22207/JPAM.16.1.50

Comment 7. Provide information on the key parameters optimized during the LAH extraction process and the optimized extraction conditions.

Response: Thank you for your comment. In this study, the parameters in the ultrasound-assisted extraction process of LAH, including extraction temperature, extraction time, extraction power, material-liquid ratio, and ethanol concentration in the solvent, were screened using a single factor experiment with the yield of total flavonoids of LAH as an index. Among them, the results of the single factor experiment for extraction time showed that the extraction time of six levels had little effect on the extraction rate of total flavonoids from LAH. While the effects of the other four factors on the extraction rate of LAH total flavonoids varied significantly at different levels, therefore, extraction temperature, extraction power, material-solvent ratio, and ethanol concentration in the solvent were selected as the key parameters for optimization of LAH extraction in the final experiment.

The results of the single factor experiment are shown in Figure 1 in the manuscript. With only one variable, the yield of total flavonoids of LAH was first increased and then decreased when the ethanol concentration was increased, and peaked at 70% ethanol concentration, so the ethanol concentration in the extraction solvent was chosen to be 60% - 80% for the subsequent experiments (Figure 1. B). Similarly, the material-solvent ratio (Figure 1. C), extraction temperature (Figure 1. D) and extraction power (Figure 1. D) were also selected for subsequent experiments at three levels centered on the peak value.

Four factors including ethanol concentration (A), material-solvent ratio (B), extraction temperature (C) and extraction power (D) were selected as variables according to the single factor experimental results. A response surface experiment with four factors and three levels was designed using Design-Expert 8.0.6 software, with -1, 0 and 1 representing variable levels. The true values and coding levels of independent variables are shown in Table 1. The experimental protocol and the corresponding results obtained by the response surface design are shown in Table 2. The quadratic multiple regression equations obtained from the subsequent analysis of the interactions between the yield of total flavonoids of LAH and the four factors are as follows:

y = 0.37 + 3.958·10-3·A – 0.02B + 0.014C + 6.867·10-3·D – 0.017AB + 1.925·10-3·AC + 0.011AD – 5.325·10-3·BC + 0.013BD + 3.825·10-3·CD – 0.031A2 – 0.035B2 – 0.025C2 – 0.023D2

where Y is the yield of total flavonoids of LAH, A is the ethanol concentration, B is the material-solvent ratio, C is the extraction temperature, and D is the extraction power.

Table. 1 Experimental factors and levels

Levels

Factors

A. Ethanol concentration (%)

B. Material-solvent ratio (g/mL)

C. Extraction temperature (°C)

D. Extraction power (W)

1

80

1: 50

80

80

0

70

1: 40

70

70

-1

60

1: 30

60

60

Table 2. Experimental design and the yield of total flavonoids of LAH

No.

Factors

Extraction rate (%)

A. Ethanol concentration

B. Material-to-solvent ratio

C. Extraction temperature

D. Extraction power

1

-1

1

0

0

0.2957

2

0

0

0

0

0.3781

3

-1

0

1

0

0.3229

4

0

-1

1

0

0.3543

5

0

0

1

1

0.3475

6

1

0

1

0

0.3314

7

1

1

0

0

0.2720

8

1

0

0

1

0.3441

9

1

0

0

-1

0.3093

10

0

0

0

0

0.3653

11

-1

0

-1

0

0.3051

12

0

1

0

-1

0.2737

13

0

0

1

-1

0.3424

14

-1

-1

0

0

0.3119

15

-1

0

0

-1

0.3220

16

0

0

0

0

0.3789

17

0

0

-1

-1

0.3127

18

-1

0

0

1

0.3119

19

0

1

1

0

0.2974

20

0

-1

0

-1

0.3237

21

0

0

-1

1

0.3025

22

0

-1

0

1

0.3297

23

1

0

-1

0

0.3059

24

0

0

0

0

0.3764

25

0

1

0

1

0.3305

26

0

-1

-1

0

0.3212

27

0

0

0

0

0.3721

28

0

1

-1

0

0.2856

29

1

-1

0

0

0.3543

The results of variance analysis of the response surface model (Table 4 in the manuscript) showed that the model has statistical significance (P < 0.05) and low error (p > 0.05 for Lack of Fit), indicating that the model has been successfully constructed and can be used to analyze the ultrasound assisted extraction process of LAH. Based on the F-value, it can be seen that the most influential factor on the yield of total flavonoids of LAH was the material-solvent ratio, which had significant influence (P < 0.05), followed by ultrasonication temperature and power, and the least influential factor was the concentration of ethanol.

According to the response surface modeling and contour plots of pairwise interaction of different factors on the yield of total flavonoids of LAH (Figure 2 in the manuscript), it can be seen that the difference in the effect of the three factors, ethanol concentration, extraction temperature and extraction power on the yield of total flavonoids of LAH when they interacted two by two was not significant. The trends of LAH total flavonoid extraction rate were all increasing then decreasing with the increment of variables. When the extraction power was in the range of 70 - 80 W and the extraction temperature was in the range of 70 - 80 °C, the contour lines were more intensive and the variation of LAH total flavonoid extraction rate was larger. When the material-solvent ratio was the variable, the interactions between it and the other three factors all had significant differences on the yield of total flavonoids of LAH. Moreover, the contour lines were very dense when the material-solvent ratio was 1: 35 - 1: 45, indicating that when the material-solvent ratio was in this interval, the yield of total flavonoids of LAH was most affected.

The analysis results of Box-Behnken software showed that the optimal extraction conditions of LAH were ethanol concentration of 71.85%, material-solvent ratio of 1: 36.73 g/mL, extraction temperature of 73.18 °C, extraction power of 71.32 W, and the predicted extraction yield of total flavonoids of LAH was 0.3804%. According to the actual operating conditions, the process conditions were changed to ethanol concentration of 72%, material-solvent ratio of 1: 37 g/mL, extraction temperature of 73 °C, extraction power of 70 W. The results of the validation experiments are shown in Table 5 in the manuscript, the average extraction yield of total flavonoids of LAH obtained by the optimized LAH extraction process was 0.3776%, which was little different from the model predicted value, and the RSD was 3.277%. The RSD value was less than 5%, indicating that the extraction conditions optimized by response surface method were accurate and reliable.

In conclusion, the key parameters optimized for LAH extraction include extraction temperature, extraction power, material-solvent ratio, and ethanol concentration in the solvent. The optimized extraction conditions by response surface methodology were ethanol concentration of 72%, material-solvent ratio of 1: 37 g/mL, extraction temperature of 73 °C, extraction power of 70 W. Thank you!

Comment 8. Explain the criteria used to select the 96 active ingredients of LAH for further study.

Response: Thank you for your comment. There are 2 selection criteria for the active ingredients of LAH. The first criterion was that the ingredient had the oral bioavailability (OB) > 30%, the drug-likeness (DL) value > 0.18, and the relative content > 1×106. According to this criterion, 74 active ingredients were screened.

OB represents the orally administered percentage of a drug which reaches the site of action without changes. Good oral bioavailability is one of the key pharmacokinetic parameters for screening new drug candidates since oral route is the most convenient and predominant mode of administration [1]. Therefore, we chose OB > 30% to screen active ingredients of LAH. DL is the index which represents the balance between pharmacodynamics affected by molecular properties and pharmacokinetic of molecules, which is used to estimate the degree of “drug-like” of the candidate components. The average DL index in Drugbank database is 0.18, which is regarded as the selection standard for the active ingredients in the traditional Chinese medicine, hence we chose 0.18 as the threshold of DL in our research [2, 3]. Furthermore, a total of 1666 substances were identified in LAH based on the results of quasi-targeted metabolomics analysis. Among them, substances with lower content were considered to have less effect on the anti-inflammatory activity of LAH extracts. Therefore, the relative content > 1 × 106 was set as another screening condition.

The second criterion for the selection of the active ingredients of LAH was the relative content of the compound > 1×106, OB < 30% or DL < 0.18, but the ingredient has been reported by literatures to have anti-inflammatory effects. Components screened by this type of criteria were also regarded as anti-inflammatory active components of LAH because of their proven anti-inflammatory activity and high relative content in LAH extracts. According to this criterion, 22 active ingredients were screened.

In conclusion, a total of 96 active ingredients in LAH were selected for further study based on these two screening criteria. Thank you!

[1] Xu, X., Zhang, W. X., Huang, C., Li, Y., Yu, H., Wang, Y. H., Duan, J. Y., Ling, Y. (2012). A novel chemometric method for the prediction of human oral bioavailability. Int. J. Mol. Sci. 13, 6964-6982. doi: 10.3390/ijms13066964

[2] Tao, W. Y., Xu, X., Wang, X., Li, B. H., Wang, Y. H., Li, Y., Yang, L. (2015). Network pharmacology-based prediction of the active ingredients and potential targets of Chinese herbal Radix Curcumae formula for application to cardiovascular disease. J. Ethnopharmacol. 145, 1-10. doi:10.1016/j.jep.2012.09.051

[3] Liu, H., Wang, J., Zhou, W., Wang, Y. H., Yang, L. (2013). Systems approaches and polypharmacology for drug discovery from herbal medicines: An example using licorice. J. Ethnopharmacol. 146, 773-793. doi: 10.1016/j.jep.2013.02.004

Comment 9. Inflammatory Response Mechanisms: Elaborate on the roles of AKT, RELA, and PTGS2 in the inflammatory response and discuss how LAH influences their expression.

Response: Thank you for your comment and suggestion. AKT (AKT1, AKT2, and AKT3) is a serine/threonine protein kinase, also known as protein kinase B (PKB), which can be activated by a variety of extracellular stimuli to play an important role in cellular responses including cell survival, metabolism, differentiation, and proliferation [1, 2]. It has been reported that AKT1 and AKT2 are involved in inflammatory responses in various tissues and diseases [2]. Among them, AKT1 is involved in pro-inflammatory signaling in macrophages and regulates macrophage innate immune function by mediating the generation of mitochondrial H2O2, its activation is associated with inflammation, oxidative stress, and accumulation of oxidized lipids [1-3]. AKT regulates the release of inflammatory factors and has been shown to have an important role in a variety of disease models, such as hepatic ischemia-reperfusion injury, acute lung injury, and hypoxic-ischemic encephalopathy [4, 5]. Meng et al. investigated the anti-neuroinflammatory effects and mechanisms of Evodiamine (EV) by establishing a model of LPS-induced inflammation of BV-1 cells, and showed that EV alleviated the neuroinflammatory response by affecting LPS-induced AKT phosphorylation [6]. In addition, in vivo and in vitro experiments have shown that Betulin, a natural product, plays an anti-inflammatory role by regulating the AKT signaling pathway to enhance the expression of anti-inflammatory genes in the nucleus and inhibit the expression and secretion of pro-inflammatory cytokines [7]. Consistent with the above studies, our results showed that LAH significantly inhibited the protein expression of AKT in the LPS-induced macrophage inflammation model in all dose groups (P < 0.01). Thus, the alleviating effect of LAH on macrophage inflammatory response is related to its blocking of the AKT signaling pathway.

RELA (also known as p65 NF-κB) is one of the subunits of the nuclear transcription factor NF-κB complex, which plays an important regulatory role in inflammatory processes [8]. Activated RELA translocates to the nucleus and acts as a transcription factor that initiates inflammatory genes to promote the production of various cytokines and chemokines in inflammatory states [9]. Therefore, RELA signaling pathway has been identified as a major contributor to the inflammatory response and may serve as a key therapeutic target for inflammatory diseases [10]. Activated AKT affects the activity of nuclear transcription factor inhibitor kinase through direct or indirect pathways thereby regulating RELA activation, nuclear translocation and RELA-dependent transcription of other genes. In addition, direct promotion of nuclear transcription factor inhibitor phosphorylation is also one of the pathways by which AKT activates RELA [11-13]. It has been reported that under inflammatory conditions, overactivation of AKT1 leads to the massive production of pro-inflammatory cytokines, which aggravates the inflammatory response [3, 14]. The LPS-induced inflammatory response in macrophages is also enhanced by the upregulation of the NF-κB signaling pathway by activated AKT [15]. Therefore, inhibition of AKT and RELA signaling is critical for the improvement of pathophysiological conditions including cancer, neuroinflammation and diabetes [16, 17]. In the study of the pharmacological effects and mechanisms of the mitochondria-targeted antioxidant N, N'-Bis(salicylideneamino)ethane-manganese (II)-8 (EUK-8), Jayachandra et al. found that it is not only a powerful antioxidant but also a multi-targeted inflammatory pathway inhibitory molecule. In addition to inhibiting pro-inflammatory cytokines and chemokines in edematous tissues, EUK-8 also could control the progression of inflammation by down-regulating the protein expression levels of AKT and RELA [18]. Similarly, the results of our experiments showed that LAH significantly inhibited the protein expression level of RELA (P < 0.01) with a certain dose-dependent trend. Therefore, the anti-inflammatory role played by LAH in the development of LPS-induced macrophage inflammation may be related to its regulation of AKT and RELA.

PTGS2 (also known as COX-2) is an inducible enzyme not expressed in normal cells, it acts as a major pro-inflammatory protein and is essential for the synthesis of the prostaglandin associated with pathological processes, especially acute and chronic inflammation [10, 19]. The high expression of PTGS2 can be detected when immune cells are stimulated by pro-inflammatory cytokines, LPS and carcinogens. Studies have shown that AKT1 plays a key role as a regulator in LPS-induced PTGS2 expression [20]. AKT1 promotes the translocation of the transcription factor NF-κB from the cytoplasm to the nucleus through the activation of the NF-κB pathway, thereby regulating the expression of inflammatory mediators such as PTGS2, iNOS, IL-1β, IL-6 and TNF-α [6]. In view of the important roles of AKT, RELA and PTGS2 in the inflammatory response process, Fang et al. investigated their roles in the alleviation of LPS-stimulated macrophage inflammation by Plantanone C (PC). The results showed that PC alleviated LPS-induced macrophage inflammation by blocking the AKT signaling pathway and attenuating the expression of LPS-activated RELA and PTGS2, thereby inhibiting the massive secretion of inflammatory mediators and proinflammatory cytokines [21]. Consistent with the above studies, our study showed that RAW264.7 cells secreted a large number of pro-inflammatory cytokines after LPS treatment, and the protein expression levels of AKT1, RELA and PTGS2 were significantly increased (P < 0.05). However, each dose of LAH could inhibit their protein expression levels to varying degrees (P < 0.05), and inhibit the secretion of pro-inflammatory cytokines at the same time. In summary, LAH may regulate the expression of pro-inflammatory mediators and the secretion of pro-inflammatory cytokines through the AKT1/RELA/PTGS2 signaling pathway, thereby alleviating LPS-induced inflammatory injury in macrophages.

We have revised the relevant parts of the manuscript based on your suggestions to more clearly elaborate the roles of AKT, RELA and PTGS2 in the inflammatory response and to discuss how LAH affects their expression (pages 21-23, lines 1008-1092). Thank you!

[1] West, A. P., Brodsky, I. E., Rahner, C., Woo, D. K., Erdjument-Bromage, H., Tempst, P., Walsh, M. C., Choi, Y., Shadel, G. S., Ghosh, S. (2011). TLR signalling augments macrophage bactericidal activity through mitochondrial ROS. Nature. 472(7344), 476-480. doi: 10.1038/nature09973

[2] Reyes-Gordillo, K., Shah, R., Arellanes-Robledo, J., Cheng, Y., Ibrahim, J., Tuma, P. L. (2019). Akt1 and akt2 isoforms play distinct roles in regulating the development of inflammation and fibrosis associated with alcoholic liver disease. Cells. 8(11), 1337. doi: 10.3390/cells8111337

[3] Kerr, B. A., Ma, L., West, X. Z., Ding, L., Malinin, N. L., Weber, M. E., Tischenko, M., Goc, A., Somanath, P. R., Penn, M. S., Podrez, E. A., Byzova, T. V., 2013. Interference with akt signaling protects against myocardial infarction and death by limiting the consequences of oxidative stress. Sci. Signal. 6(287), ra67. doi: 10.1126/scisignal.2003948

[4] Chen, G., Liu, J., Jiang, L., Ran, X., He, D., Li, Y., Huang, B., Wang, W., Fu, S. (2017). Galangin Reduces the Loss of Dopaminergic Neurons in an LPS-Evoked Model of Parkinson's Disease in Rats. Int J Mol Sci. 19(1), 12. doi: 10.3390/ijms19010012

[5] Yan, J., Li, J., Zhang, L., Sun, Y., Jiang, J., Huang, Y., Xu, H., Jiang, H., Hu, R. (2018). Nrf2 protects against acute lung injury and inflammation by modulating TLR4 and Akt signaling. Free Radic Biol Med. 121, 78-85. doi: 10.1016/j.freeradbiomed.2018.04.557

[6] Meng, T., Fu, S., He, D., Hu, G., Gao, X., Zhang, Y., Huang, B., Du, J., Zhou, A., Su, Y., Liu, D. (2021). Evodiamine Inhibits Lipopolysaccharide (LPS)-Induced Inflammation in BV-2 Cells via Regulating AKT/Nrf2-HO-1/NF-κB Signaling Axis. Cell Mol Neurobiol. 41(1), 115-127. doi: 10.1007/s10571-020-00839-w

[7]   Ren, C., Jin, J., Hu, W., Chen, Q., Yang, J., Wu, Y., Zhou, Y., Sun, L., Gao, W., Zhang, X., Tian, N. (2021). Betulin Alleviates the Inflammatory Response in Mouse Chondrocytes and Ameliorates Osteoarthritis via AKT/Nrf2/HO-1/NF-κB Axis. Front Pharmacol. 12, 754038. doi: 10.3389/fphar.2021.754038

[8] Tang, B., Li, X., Ren, Y., Wang, J., Xu, D., Hang, Y., Zhou, T., Li, F., Wang, L. (2017). MicroRNA-29a regulates lipopolysaccharide (LPS)-induced inflammatory responses in murine macrophages through the Akt1/ NF-κB pathway. Exp. Cell Res. 360(2), 74-80. doi: 10.1016/j.yexcr.2017.08.013

[9] Lawrence, T. (2009). The nuclear factor NF-kappaB pathway in inflammation. Cold Spring Harb Perspect Biol. 1(6), a001651. doi: 10.1101/cshperspect.a001651 84.

[10] Wang, L., Gu, J., Zong, M., Zhang, Q., Li, H., Li, D., Mou, X., Liu, P., Liu, Y., Qiu, F., Zhao, F. (2021). Anti-inflammatory action of physalin A by blocking the activation of NF-κB signaling pathway. J Ethnopharmacol. 267, 113490. doi: 10.1016/j.jep.2020.113490

[11] Hideshima, T., Chauhan, D., Richardson, P., Mitsiades, C., Mitsiades, N., Hayashi, T., Munshi, N., Dang, L., Castro, A., Palombella, V., Adams, J., Anderson, K. C. (2002). NF-kappa B as a therapeutic target in multiple myeloma. J. Biol. Chem. 277(19), 16639. doi: 10.1074/jbc.M200360200

[12] Vandermoere, F., El Yazidi-Belkoura, I., Adriaenssens, E., Lemoine, J., Hondermarck, H. (2005). The antiapoptotic effect of fibroblast growth factor-2 is mediated through nuclear factor-κB activation induced via interaction between Akt and IκB kinase-β in breast cancer cells. Oncogene. 24(35), 5482-5491. doi: 10.1038/sj.onc.1208713

[13] Idris, A. I., Libouban, H., Nyangoga, H., Landao-Bassonga, E., Chappard, D., Ralston, S. H. (2009). Pharmacologic inhibitors of I kappa B kinase suppress growth and migration of mammary carcinosarcoma cells in vitro and pre-vent osteolytic bone metastasis in vivo. Mol. Cancer Ther. 8(8), 2339. doi: 10.1158/1535-7163.MCT-09-0133

[14] Cubero, F. J., Nieto, N. (2006). Kupffer cells and alcoholic liver disease. Rev. Esp. De Enferm. Dig. Organo De La Soc. Esp. De Patol. Dig. 98(6), 460-472. doi: 10.4321/S1130-01082006000600007

[15] Wan, J., Shan, Y., Fan, Y., Fan, C., Chen, S., Sun, J., Zhu, L., Qin, L., Yu, M., Lin, Z. (2016). NF-kappaB inhibition at-tenuates LPS-induced TLR4 activation in monocyte cells. Mol. Med. Rep. 14 (5), 4505-4510. doi: 10.3892/mmr.2016.5825

[16] Hu, X., Zhang, H., Zhuang, L., Jin, G., Yang, Q., Li, M., Sun, W., Chen, F. (2020). Ubiquitin-Fold Modifier-1 Participates in the Diabetic Inflammatory Response by Regulating NF-κB p65 Nuclear Translocation and the Ubiquitination and Degradation of IκBα. Drug Des Devel Ther. 14, 795-810. doi: 10.2147/DDDT.S238695

[17] Hour, M. J., Tsai, S. C., Wu, H. C., Lin, M. W., Chung, J. G., Wu, J. B., Chiang, J. H., Tsuzuki, M., Yang, J. S. (2012). Anti-tumor effects of the novel quinazolinone MJ-33: inhibition of metastasis through the MAPK, AKT, NF-κB and AP-1 signaling pathways in DU145 human prostate cancer cells. Int J Oncol. 41(4), 1513-9. doi: 10.3892/ijo.2012.1560

[18] Jayachandra, K., Gowda, M. D. M., Rudresha, G. V., Manjuprasanna, V. N., Urs, A. P., Nandana, M. B., Bharatha, M., Jameel, N. M., Vishwanath, B. S. (2023). Inhibition of sPLA2 enzyme activity by cell-permeable antioxidant EUK-8 and downregulation of p38, Akt, and p65 signals induced by sPLA2 in inflammatory mouse paw edema model. J Cell Biochem. 124(2), 294-307. doi: 10.1002/jcb.30366

[19] Linghu, K. G., Ma, Q. S., Zhao, G. D., Xiong, W., Lin, L., Zhang, Q. W., Bian, Z., Wang, Y., Yu, H. (2020). Leocarpinolide B attenuates LPS-induced inflammation on RAW264.7 macrophages by mediating NF-κB and Nrf2 pathways. Eur J Pharmacol. 868, 172854. doi: 10.1016/j.ejphar.2019.172854

[20] Li, Y., Zou, L., Li, T., Lai, D., Wu, Y., Qin, S. (2019). Mogroside V inhibits LPS-induced COX-2 expression/ROS pro-duction and overexpression of HO-1 by blocking phosphorylation of AKT1 in RAW264.7 cells. Acta Biochim Biophys Sin (Shanghai). 51(4), 365-374. doi: 10.1093/abbs/gmz014

[21] Fang, Y., Yang, L., He, J. (2021). Plantanone C attenuates LPS-stimulated inflammation by inhibiting NF-κB/iNOS/COX-2/MAPKs/Akt pathways in RAW 264.7 macrophages. Biomed Pharmacother. 143, 112104. doi: 10.1016/j.biopha.2021.112104

Comment 10. Provide additional information on the role of MAPKs and JUN in mediating the production of inflammatory mediators and detail how LAH affects their expression.

Response: Thank you for your comment and suggestion. Accumulated evidence suggests that the upregulation of the activity of non-receptor tyrosine kinases, including AKT, SYK, SRC, JAK, PI3K, and MAPKs, is also involved in mediating the overproduction of inflammatory mediators [1]. Among them, MAPKs consist of three isoforms, JNK, MAPK3 (ERK1), and p38 kinase, which are members of the highly conserved serine/threonine protein kinase family [2]. As known up-regulators of NF-κB, MAPKs play a critical control role in processes associated with cellular transduction including inflammation, cancer metastasis and cell proliferation [3]. Phosphorylated MAPKs binds and activates target kinases and then translocate to the nucleus to further activate the transcription of pro-inflammatory genes. It has been reported that MAPKs activate intracellular signal transduction in the NF-κB pathway, thereby inducing the expression of inflammatory mediators such as IL-6 and TNF-α [4]. In addition, MAPKs and NF-κB signaling pathways have also been demonstrated to be involved in TNF-α stimulation process of several cytokines that contribute to the various inflammatory pathogenesis [5]. It has been shown that signaling cascades including MAPKs and AKT are activated upon LPS binding to TLR4 receptor, which directly promotes DNA-binding activity in macrophages by phosphorylating transcription factors and activating downstream signaling pathways such as NF-κB and AP-1, thereby upregulating the mRNA expression levels of inflammatory mediators such as TNF-α and iNOS [6-8]. AP-1 transcription factor, a downstream target of MAPKs, is a homologous or heterodimeric protein complex composed of the FOS (c-Fos, FosB, Fra-1 and Fra-2) and JUN (c-Jun, JunB and JunD) subfamilies [9, 10]. As regulators of the AP-1 transcription factor, activated MAPKs stimulate the activation of the transcription factor JUN and nuclear translocation, a process that has also been shown to be associated with pro-inflammatory gene expression [11]. The study of MAPK/AP-1 signaling in airway epithelial cells have revealed that activation of TLR is associated with this signaling pathway and may contribute to the production of inflammatory mediators and host defense proteins [12]. Therefore, MAPKs regulate the production of pro-inflammatory mediators and cytokines and influence cellular responses to environmental stresses by activating AP-1 [13, 14]. Studies have shown that AP-1 plays a critical mediated role as the key transcription factor in the transcriptional activation of multiple inflammatory genes [15]. Specifically, AP-1 is located in the cytoplasm under normal physiological conditions and is induced to translocate to the nucleus upon activation of MAPKs by inflammatory stimuli such as LPS. The inflammatory signaling cascade is thus initiated, which in turn affects the expression of inflammatory mediators PTGS2, iNOS, and pro-inflammatory cytokines IL-1β, IL-6, and TNF-α [16]. In addition, it has been reported that PTGS2 expression can be post-transcriptionally regulated by MAPKs through activation of AP-1 complexes or mRNA stabilization [17].

Mitra et al. demonstrated through in vivo and in vitro experiments that Korean Red Ginseng water extract had significant inhibitory effect on cadmium-induced pro-inflammatory cytokine production by inactivating the MAPKs/AP-1 signaling path-way [18]. Sakai et al. reported that astaxanthin could blocked DSS-induced translocation of RELA and JUN into the nucleus of mucosal epithelial cells, while inhibiting the activation of MAPKs in mucosal, thereby significantly reducing the mRNA expression levels of the pro-inflammatory cytokines IL-1β, IL-6, and TNF-α, and exerting a mitigating effect on inflammatory bowel disease [16]. Moreover, Abekura et al. showed that inhibition of MAPKs, NF-κB, and AP-1 signaling pathways significantly down-regulated the expression levels of pro-inflammatory cytokines and inflammatory mediators (PTGS2, NO), and thus suppressed LPS-induced inflammatory responses in murine macrophages [19]. Our findings are consistent with those content reported above that the protein expression levels of MAPK3 and JUN were significantly elevated in LPS-stimulated macrophages compared with the blank control group (P < 0.01), whereas LAH significantly reversed this trend (P < 0.01. At the same time, the secretion levels of pro-inflammatory cytokines and the protein expression levels of the pro-inflammatory mediator PTGS2 in all three LAH dosage groups were significantly down-regulated. Thus, our data strongly suggested that LAH may have a role in alleviating macrophage inflammation by inhibiting the activation of the LPS-induced MAPK3/JUN signaling pathway, while downregulating the levels of pro-inflammatory cytokines and inflammatory mediators.

We have revised relevant part in the manuscript (pages 23-24, lines 1093-1165) according your advice. Thank you!

[1] Hu, W., Wang, X., Wu, L., Shen, T., Ji, L., Zhao, X., Si, C. L., Jiang, Y., Wang, G. (2016). Apigenin-7-O-β-D-glucuronide inhibits LPS-induced inflammation through the inactivation of AP-1 and MAPK signaling pathways in RAW 264.7 macrophages and protects mice against endotoxin shock. Food Funct. 7(2), 1002-1013. doi: 10.1039/c5fo01212k

[2] Tang, Z. P., Cui, Q. Z., Dong, Q. Z., Xu, K., Wang, E. H. (2013). Ataxia–telangiectasia group D complementing gene (ATDC) upregulates matrix metalloproteinase 9 (MMP-9) to promote lung cancer cell invasion by activating ERK and JNK pathways. Tumour Biol. 34, 2835-2842. doi: 10.1007/s13277-013-0843-7

[3] LoBue, P. A., Enarson, D. A., Thoen, T. C. (2010). Tuberculosis in humans and its epidemiology, diagnosis and treatment in the United States. Int J Tuberc Lung Dis. 14(10), 1226-32. doi: 10.3109/01902141003721457

[4]   Nishida, A., Hidaka, K., Kanda, T., Imaeda, H., Shioya, M., Inatomi, O., Bamba, S., Kitoh, K., Sugimoto, M., Andoh, A. (2016). Increased Expression of Interleukin-36, a Member of the Interleukin-1 Cytokine Family, in Inflammatory Bowel Disease. Inflamm Bowel Dis. 22(2), 303-14. doi: 10.1097/MIB.0000000000000654

[5]   Hayden, M. S., Ghosh, S. (2014). Regulation of NF-κB by TNF family cytokines. Semin Immunol. 26(3), 253-66. doi: 10.1016/j.smim.2014.05.004

[6]   Ridnour, L. A., Cheng, R. Y., Switzer, C. H., Heinecke, J. L., Ambs, S., Glynn, S. (2013). Molecular pathways: Toll-like receptors in the tumor microenvironment-poor prognosis or new therapeutic opportunity. Clin. Cancer Res. 19, 1340-1346. doi: 10.1158/1078-0432.CCR-12-0408

[7]   Endale, M., Park, S. C., Kim, S., Kim, S. H., Yang, Y., Cho, J. Y. (2013). Quercetin disrupts tyrosine-phosphorylated phosphatidylinositol 3-kinase and myeloid differentiation factor-88 association, and inhibits MAPK/AP-1 and IKK/NF-κB-induced inflammatory mediators production in RAW 264.7 cells. Immunobiology. 218, 1452-1467. doi: 10.1016/j.imbio.2013.04.019

[8]   Chun, J., Choi, R. J., Khan, S., Lee, D. S., Kim, Y. C., Nam, Y. J. (2012). Alantolactone suppresses inducible nitric oxide synthase and cyclooxygenase-2 expression by down-regulating NF-κB, MAPK and AP-1 via the MyD88 signaling pathway in LPS-activated RAW 264.7 cells, Int. Immunopharmacol. 14, 375-383. doi: 10.1016/j.intimp.2012.08.011

[9]   Yang, S. H., Sharrocks A. D., Whitmarsh, A. J. (2013). MAP kinase signalling cascades and transcriptional regulation. Gene. 513, 1-13. doi: 10.1016/j.gene.2012.10.033

[10] Hess, J., Angel, P., Schorpp-Kistner, M. (2004). AP-1 subunits: quarrel and harmony among siblings. J. Cell Sci. 117, 5965-5973. doi: 10.1242/jcs.01589

[11] Hossen, M. J., Kim, M. Y., Cho, J. Y. (2016). MAPK/AP-1-Targeted Anti-Inflammatory Activities of Xanthium strumarium. Am J Chin Med. 44(6), 1111-1125. doi: 10.1142/S0192415X16500622

[12] Thaikoottathil, J., Chu, H. W. (2011). MAPK/AP-1 activation mediates TLR2 agonist-induced SPLUNC1 expression in human lung epithelial cells. Mol Immunol. 49(3), 415-22. doi: 10.1016/j.molimm.2011.08.005

[13] Kuo, M. Y., Liao, M. F., Chen, F. L., Li, Y. C., Yang, M. L., Lin, R. H., Kuan, Y. H. (2011). Luteolin attenuates the pulmonary inflammatory response involves abilities of antioxidation and inhibition of MAPK and NFκB pathways in mice with endotoxin-induced acute lung injury. Food Chem. Toxicol. 49(10), 2660-2666. doi: 10.1016/j.fct.2011.07.012

[14] Khan, A., Khan, S., Ali, H., Shah, K. U., Ali, H., Shehzad, O., Onder, A., Kim, Y. S. (2019). Anomalin attenuates LPS-induced acute lungs injury through inhibition of AP-1 signaling. Int Immunopharmacol. 73, 451-460. doi: 10.1016/j.intimp.2019.05.032

[15] Patil, R. H., Naveen Kumar, M., Kiran Kumar, K. M., Nagesh, R., Kavya, K., Babu, R. L., Ramesh, G. T., Chidananda Sharma, S. (2017). Dexamethasone inhibits inflammatory response via down regulation of AP-1 transcription factor in human lung epithelial cells. Gene. 645, 85-94. doi: 10.1016/j.gene.2017.12.024

[16] Sakai, S., Nishida, A., Ohno, M., Inatomi, O., Bamba, S., Sugimoto, M., Kawahara, M., Andoh, A. (2019). Astaxanthin, a xanthophyll carotenoid, prevents development of dextran sulphate sodium-induced murine colitis. J Clin Biochem Nutr. 64(1), 66-72. doi: 10.3164/jcbn.18-47

[17] Looby, E., Abdel-Latif, M. M., Athié-Morales, V., Duggan, S., Long, A., Kelleher, D. (2009). Deoxycholate induces COX-2 expression via Erk1/2-, p38-MAPK and AP-1-dependent mechanisms in esophageal cancer cells. BMC Cancer. 9, 190. doi: 10.1186/1471-2407-9-190

[18] Mitra, A., Rahmawati, L., Lee, H. P., Kim, S. A., Han, C. K., Hyun, S. H., Cho, J. Y. (2022). Korean Red Ginseng water extract inhibits cadmium-induced lung injury via suppressing MAPK/ERK1/2/AP-1 pathway. J Ginseng Res. 46(5), 690-699. doi: 10.1016/j.jgr.2022.04.003

[19] Abekura, F., Park, J., Lim, H., Kim, H. D., Choi, H., Lee, M. J., Kim, C. H. (2022). Mycobacterium tuberculosis glycoli-poprotein LprG inhibits inflammation through NF-κB signaling of ERK1/2 and JNK in LPS-induced murine macrophage cells. J Cell Biochem. 123(4), 772-781. doi: 10.1002/jcb.30220

Comment 11. Discussion: The discussion section is weakly written and would benefit from incorporating more recent updates.

Response: Thank you for your comment and suggestion. We apologize for the deficiencies in writing of the discussion section. We have substantially revised the discussion section in the manuscript (pages 25-32, lines 690-1015) and added more new relevant literature for a more in-depth and comprehensive discussion of this study. Thank you again for your valuable comments and constructive suggestions. Thank you!

Comment 12. Graphical Abstract: If possible, consider providing a graphical abstract to enhance the visual representation of your research.

Response: Thank you for your comment and suggestion. We have added the graphical abstract in the manuscript (page 4, lines 149-150) according your advice. Thank you!

Comment 13. Conclusion: In light of some challenges encountered in reading and comprehending certain parts of the manuscript, critical corrections are needed. However, the manuscript addresses important issues, employs interesting approaches and techniques, and contributes to our understanding of LAH's role in anti-inflammatory activity and related pathways. With some minor revisions, I believe this manuscript is suitable for publication in IJMS.

Response: Thank you again for your pertinent comments and constructive suggestions, which are crucial to improving the quality of our manuscript. Your affirmation gives us great encouragement. We have carefully revised the manuscript according to each of your comments and suggestions. Thank you very much!

Reviewer 2 Report

Comments and Suggestions for Authors

The manuscript entitled, ‘Study on the alleviating effect and potential mechanism of Limonium aureum on LPS-induced inflammatory response in macrophages’ examines the anti-inflammatory effects of ethanolic extract of Limonium aureum and the probable mechanism behind using network pharmacology and molecular biology. The findings of the anti-inflammatory activity of the extract on RAW264.7 cells are interesting. However, the manuscript lacks the presentation and writing part, some major concerns should be addressed by the authors before this manuscript can be recommended for publication in the International Journal of Molecular Sciences.

1.      The #Abstract section sounds like an introduction; the #Abstract should briefly overview the whole work. It should be rewritten, including some important findings and the conclusion of this study.

2.      Since ethanol is the organic solvent used for the extraction of metabolites, the #title of this study must include that; authors may write, ‘…...mechanism of ethanolic extract of Limonium aureum on LPS….’.

3.      The quality of #Figure_1,3 and 5 must be improved, and the size of the legends should be corrected for better representation.

4.      The authors have not presented any in-vitro experimental findings. On what basis the authors have chosen to perform the anti-inflammatory assay on RAW cells?

5.      The cell viability analysis (#Figure_3A) indicates that certain concentrations (20-40 microgram/mL) of the treatment augment the viability of RAW264.7 cells ~150%. What is the possible reason behind this? It should be explained in the respective results and discussion section.

6.      The statistical significance should be explained comprehensively in the figures’ descriptions.

7.      What are the major compounds present in the LAH that are responsible for the anti-inflammatory activity of the extract?

8.      The authors must prepare suitable figures for scientific presentations; for instance, the data provided in the supplementary materials must be compiled in a single supplementary file.

9.      The authors may compare the findings of this study with previously reported similar literature to improve the quality of this manuscript.

10.  The whole manuscript should be thoroughly checked for English language and grammatical errors.

Comments on the Quality of English Language

The manuscript should be thoroughly checked for the English language.

Author Response

Response to Reviewer 2:

Dear Reviewer,

Thank you very much for your comments and suggestion concerning our manuscript. The comments and suggestions are valuable and very helpful for revising and improving our paper, as well as the important guiding significance to our researches. We have taken all these comments and suggestions into account, and have made major corrections in this revised manuscript. The responses answering every question are as follows. All the lines and pages mentioned in responses are form the revised manuscript in review mode (Word version).

Comments and Suggestions.

The manuscript entitled, ‘Study on the alleviating effect and potential mechanism of Limonium aureum on LPS-induced inflammatory response in macrophages’ examines the anti-inflammatory effects of ethanolic extract of Limonium aureum and the probable mechanism behind using network pharmacology and molecular biology. The findings of the anti-inflammatory activity of the extract on RAW264.7 cells are interesting. However, the manuscript lacks the presentation and writing part, some major concerns should be addressed by the authors before this manuscript can be recommended for publication in the International Journal of Molecular Sciences.

Response: Thank you for your pertinent comments and constructive suggestions. Your affirmation gives us great encouragement. We have carefully revised the manuscript based on each comment and suggestion you pointed. Thank you!

Comment 1. The #Abstract section sounds like an introduction; the #Abstract should briefly overview the whole work. It should be rewritten, including some important findings and the conclusion of this study.

Response: Thank you for pointing this out. We apologize for the inappropriate description. We have revised the abstract in the manuscript (page 1, lines 15-41) according your advice. Thank you!

Comment 2. Since ethanol is the organic solvent used for the extraction of metabolites, the #title of this study must include that; authors may write, ‘…...mechanism of ethanolic extract of Limonium aureum on LPS….’.

Response: Thank you for pointing this out. We have revised the title in the manuscript (page 1, lines 2-3) according your advice. Thank you!

Comment 3. The quality of #Figure_1,3 and 5 must be improved, and the size of the legends should be corrected for better representation.

Response: Thank you for your comment and suggestion. We are sorry for the poor quality of these figures. We have revised Figure 1, 3 and 5 in the manuscript. Thank you!

Comment 4. The authors have not presented any in-vitro experimental findings. On what basis the authors have chosen to perform the anti-inflammatory assay on RAW cells?

Response: Thank you for your comment and suggestion. In this study, the in vitro anti-inflammatory activity of LAH was evaluated by establishing the LPS-induced inflammatory macrophage model. Firstly, we used CCK-8 assay to screen the non-cytotoxic doses of LAH. Based on the results of the cytotoxicity test, LAH of 10, 30, 50 μg/mL was selected for subsequent experiments. It has been reported that inflammatory macrophages secrete pro-inflammatory cytokines IL-6, TNF-α, and IFN-γ in large amounts and further exacerbate the inflammatory response [1, 2]. Consistent with previous reports, our experimental results also showed that macrophages exposed to LPS secreted pro-inflammatory cytokines IFN-γ, IL-6, and TNF-α in large quantities, while LAH treatment significantly reversed the promotion of pro-inflammatory cytokine secretion by LPS. In contrast, secretion of anti-inflammatory cytokines such as IL-4, IL-10 and Arg-1 contribute to defense against excessive inflammation [3]. Our assay of the secretion of anti-inflammatory cytokines showed that LAH promoted the secretion of anti-inflammatory cytokines IL-4 and IL-10, and there was a significant difference (P < 0.05) compared with LPS group. Thus, LAH may alleviate LPS-induced inflammatory responses in macrophages by promoting the secretion of anti-inflammatory cytokines and inhibiting the secretion of pro-inflammatory cytokines.

Macrophages are widely existed in lymphoid tissues and are first line of immune defense against infection [4]. RAW264.7 is a cell line obtained by collecting mouse ascites mononuclear-like macrophages from murine leukemia virus-induced tumors in BALB/c mice (ATCC Number: TIB-71). This cell has a strong ability to adhere and phagocytose antigens and plays a key role in inflammatory, immune and phagocytic reactions, and is a commonly used cell line in microbiology and immunology studies [5, 6]. The inflammatory response model constructed using LPS-stimulated RAW264.7 cells was stable, and the contrast between the model group and the blank group was obvious, with significant changes in the secretion of pro-inflammatory cytokines and anti-inflammatory cytokines. Therefore, this model is often applied to carry out the evaluation of drug anti-inflammatory activity and the study of mechanism. Currently, LPS-induced RAW264.7 macrophages are widely used as a screening model in the study of anti-inflammatory activity of natural products. The effects of natural products on the secretion level of inflammation-related factors and the expression level of inflammation-related proteins in this model were analyzed by ELISA assay and Western blotting, respectively, which can effectively determine whether the natural products have potential anti-inflammatory activities. For example, orientin and vitexin have been reported to regulate the MAPK/NF-κB signaling pathway by inhibiting NF-κB translocation in RAW264.7 cells with LPS-induced inflammation, thereby inhibiting the production of inflammatory mediators and attenuating inflammatory responses [7]. By studying this model, Zou et al. found that the sesquiterpenoids contained in the traditional Chinese medicine Salvia plebeia significantly reduced the release and expression of inflammation-related factors and exerted their anti-inflammatory activities by inhibiting the NF-κB and Erk1/2 signaling pathways [8]. Therefore, in this study, RAW264.7 cells were selected to determine the in vitro anti-inflammatory activity of LAH. Thank you!

[1] Du, N., Wu, K., Zhang, J., Wang, L., Pan, X., Zhu, Y., Wu, X., Liu, J., Chen, Y., Ye, Y., Wang, Y., Wu, W., Cheng, W., Huang, Y. (2021). Inonotsuoxide B regulates M1 to M2 macrophage polarization through sirtuin-1/endoplasmic reticulum stress axis. Int Immunopharmacol. 96, 107603. doi: 10.1016/j.intimp.2021.107603

[2] Park, J. K., Shao, M., Kim, M. Y., Baik, S. K., Cho, M. Y., Utsumi, T., Satoh, A., Ouyang, X., Chung, C., Iwakiri, Y. (2017). An endoplasmic reticulum protein, Nogo-B, facilitates alcoholic liver disease through regulation of kupffer cell polarization. Hepatology. 65(5), 1720-1734. doi: 10.1002/hep.29051

[3] Liu, M., Chen, Y., Wang, S., Zhou, H., Feng, D., Wei, J., Shi, X., Wu, L., Zhang, P., Yang, H., Lv, X. (2020). α-Ketoglutarate Modulates Macrophage Polarization Through Regulation of PPARγ Transcription and mTORC1/p70S6K Pathway to Ameliorate ALI/ARDS. Shock. 53(1), 103-113. doi: 10.1097/SHK.0000000000001333. PMID: 31841452.

[4] Zhou, D., Huang, C., Lin, Z., Zhan, S., Kong, L., Fang, C., Li, J. (2014). Macrophage polarization and function with emphasis on the evolving roles of coordinated regulation of cellular signaling pathways. Cell Signal. 26(2), 192-197. doi: 10.1016/j.cellsig.2013.11.004

[5] Andreyev, A. Y., Fahy, E., Guan, Z., Kelly, S., Li, X., McDonald, J. G., Milne, S., Myers, D., Park, H., Ryan, A., Thompson, B. M., Wang, E., Zhao, Y., Brown, H. A., Merrill, A. H., Raetz, C. R., Russell, D. W., Subramaniam, S., Dennis, E. A. (2010). Subcellular organelle lipidomics in TLR-4-activated macrophages. J. Lipid Res. 51(9), 2785-2797. doi: 10.1194/jlr.M008748

[6] Lee, S. J., Lim, K. T. (2008). Phytoglycoprotein inhibits interleukin-1beta and interleukin-6 via p38 mitogen-activated protein kinase in lipopolysaccharide-stimulated RAW 264.7 cells. Naunyn Schmiedebergs Arch Pharmacol. 377(1), 45-54. doi: 10.1007/s00210-007-0253-8

[7] Yu, Y., Pei, F., Li, Z. (2022). Orientin and vitexin attenuate lipopolysaccharide-induced inflammatory responses in raw264.7 cells: a molecular docking study, biochemical characterization, and mechanism analysis. Food Sci. Hum. Well. 11(5), 1273-1281. doi: 10.1016/j.fshw.2022.04.024

[8] Zou, Y. H., Zhao, L., Xu, Y. K., Bao, J. M., Liu, X., Zhang, J. S., Li, W., Ahmed, A., Yin, S., Tang, G. H. (2018). Anti-inflammatory sesquiterpenoids from the Traditional Chinese Medicine Salvia plebeia: Regulates pro-inflammatory mediators through inhibition of NF-κB and Erk1/2 signaling pathways in LPS-induced RAW264.7 cells. J Ethnopharmacol. 210, 95-106. doi: 10.1016/j.jep.2017.08.034

Comment 5. The cell viability analysis (#Figure_3A) indicates that certain concentrations (20-40 microgram/mL) of the treatment augment the viability of RAW264.7 cells ~150%. What is the possible reason behind this? It should be explained in the respective results and discussion section.

Response: Thank you for your comment and suggestion. In this study, the effect of LAH on the cell viability of RAW264.7 cells was detected by CCK-8 method, which reflects the cell proliferation activity by detecting the concentration of succinate dehydrogenase in mitochondria in cells. Cell proliferation is one of the fundamental processes in the development of organisms and the continuation of life, and its regulation in organisms is influenced by a variety of factors and mechanisms, including growth factors, cell cycle regulation, apoptosis regulation, and oxidative metabolism regulation. The results of our experiment indicated that LAH concentrations below 100 µg/mL are non-cytotoxic, and LAH concentrations below this value can be selected as non-cytotoxic doses for subsequent experiments. In addition, it is noteworthy that 10 - 50 µg/mL of LAH significantly promoted cell proliferative activity. Studies have shown that bioactive compounds in natural products have promoting effect on the expression of growth factors associated with cell proliferation, e.g., the ethanol extract of Hericium erinaceus enhanced the synthesis of neurotrophic factor (NGF) through the JNK pathway, which promoted the proliferation of hippocampal neural stem cells [1]. It is hypothesized that the promoting effect of LAH on cell proliferation may be related to its promotion of growth factor expression. Fu et al. investigated the effects of Cibotium barometz polysaccharides (CBPS) on the viability and cell cycle transition of primary chondrocytes in rats and found that CBPS had an enhancing effect on chondrocyte proliferation, which may be related to its promotion of G1/S cell cycle transition [2]. Therefore, the regulatory effect of LAH on the cell cycle may also be involved in its promotion of cell proliferation. Furthermore, the natural product Loureirin B (LB) has been shown to promote the proliferation of Ins-1 cells while inhibiting their apoptosis [3]. It is inferred that the role of LAH in promoting cell proliferation is also related to its regulation of apoptosis.Astragalus polysaccharide (APS) has been shown to reduce the levels of ROS, MDA and NO in diabetic cardiomyopathy (DCM) cell models, improve the activity of antioxidant enzymes, and promote the proliferation of DCM cells, thus playing a protective role in DCM cells, which may be related to the activation of NGR1/ErbB signaling pathway by APS [4]. Moreover, LAH has been reported to have favorable antioxidant activity in previous studies [5], thus its modulation of oxidative metabolism may also be involved in its promotion of cell proliferation.

We have added relevant explanation to the discussion section of the manuscript (page 27, lines 768-789) according your suggestion. Thank you!

[1] Chong, P. S., Fung, M. L., Wong, K. H., Lim, L. W. (2019). Therapeutic potential of Hericium erinaceus for Depressive Disorder. Int J Mol Sci. 21(1), 163. doi: 10.3390/ijms21010163

[2] Fu, C., Zheng, C., Lin, J., Ye, J., Mei, Y., Pan, C., Wu, G., Li, X., Ye, H., Liu, X. (2017). Cibotium barometz polysaccharides stimulate chondrocyte proliferation in vitro by promoting G1/S cell cycle transition. Mol Med Rep. 15(5), 3027-3034. doi: 10.3892/mmr.2017.6412

[3] Fang, H., Ding, Y., Xia, S., Chen, Q., Niu, B. (2022). Loureirin B promotes insulin secretion through GLP-1R and AKT/PDX1 pathways. Eur J Pharmacol. 936, 175377. doi: 10.1016/j.ejphar.2022.175377

[4] Chang, X., Lu, K., Wang L, Lv M, Fu W. (2018). Astraglaus polysaccharide protects diabetic cardiomyopathy by activating NRG1/ErbB pathway. Biosci Trends. 12(2), 149-156. doi: 10.5582/bst.2018.01027

[5] Yang, Z., Mo, Y., Cheng, F., Zhang, H., Shang, R., Wang, X., Liang, J., Liu, Y., Hao, B. (2022). Antioxidant effects and potential molecular mechanism of action of Limonium aureum extract based on systematic network pharmacology. Front Vet Sci. 8, 775490. doi: 10.3389/fvets.2021.775490

Comment 6. The statistical significance should be explained comprehensively in the figures’ descriptions.

Response: Thank you for pointing this out. We apologize for our carelessness. We have added the explanation of the statistical significance in the descriptions of Figures 3 and 5 (pages 17-18, lines 529-531; page 25, lines 681-683). Thank you!

Comment 7. What are the major compounds present in the LAH that are responsible for the anti-inflammatory activity of the extract?

Response: Thank you for your comment and suggestion. In this study, the components in LAH extracts were analyzed by quasi-targeted metabolomics approach, which resulted in the identification of a total of 1666 components from LAH extracts, with specific information and relative quantitative values as shown in Table S2. We screened and further analyzed 1666 components based on their oral bioavailability, drug-likeness, relative content values and whether the component has been reported to have anti-inflammatory activity. However, our study still has some limitations. These include the fact that the specific components of LAH extracts that exert anti-inflammatory activity as a mixture are still unclear. Although quasi-targeted metabolomics analyses provided some information on the components contained in LAH, more in-depth studies are remaining needed to determine the specific components in which LAH exerts anti-inflammatory activity. The separation, purification of the extracts and application of chromatographic and spectroscopic techniques will help to further explore the specific information of anti-inflammatory active components in LAH extract. In future studies, we will use the above techniques and methods to deepen the research on LAH anti-inflammatory active ingredients in order to improve the quality and level of our research. Thank you again for your valuable comments, which will make our research work more comprehensive. Thank you!

Comment 8. The authors must prepare suitable figures for scientific presentations; for instance, the data provided in the supplementary materials must be compiled in a single supplementary file.

Response: Thank you for your comment and suggestion. We have compiled the supplementary materials into a single supplementary file according your suggestion. Thank you!

Comment 9. The authors may compare the findings of this study with previously reported similar literature to improve the quality of this manuscript.

Response: Thank you for your comment and suggestion. We have added relevant content to the discussion section of the manuscript based on your suggestion. Thank you!

Comment 10. The whole manuscript should be thoroughly checked for English language and grammatical errors.

Response: Thank you for your comment and suggestion. We apologize for some of the language and grammatical errors in the manuscript caused by the limitations of our English writing skills. We have done our best to check and revise these errors in the manuscript. Thank you!

Reviewer 3 Report

Comments and Suggestions for Authors

The authors have crafted an intriguing piece of work that undoubtedly captures the interest of readers and holds the potential to significantly advance the field of research. However, prior to publication, it is essential for the authors to attentively address the following comments to ensure the manuscript meets the highest standards of clarity, precision, and completeness.

The manuscript requires several enhancements for clarity and visual appeal.

 Firstly, in the title, it is advisable to include the full name of "Li-monium aureum."

In the Abstract section, the author should provide the full form of LAH, highlight key study findings, and underscore the significance of the work, thereby reducing the introduction's length.

Address Line 162 by questioning the rationale behind selecting specific material parameters in the experimental design, especially when these values were screened in steps 2, 3, and 4.

The figures in the manuscript demand an enhancement in quality. The font size throughout all figures is challenging to discern, particularly the X and Y labels and fonts. A crucial improvement lies in making these elements more visibly clear and reader-friendly.

For example, Figure 1 needs a revamp for better representation—increasing the size and pixel quality, with a particular focus on elevating the font size of the X and Y axes. Subfigure 1's axes are challenging to follow, demanding an improvement. In subsequent figures, there is a consistent need for increased axis label font sizes to enhance legibility. Additionally, Figure 4 requires attention, as the current font size impedes clear visibility.

Considering LAH is a mixture of flavonoids, the author should include the percentage composition of each component rather than relying solely on quercetin for quantification. A broader spectrum, as indicated by a calibration plot covering various components, would enhance the credibility of the study.

Author Response

Response to Reviewer 3:

Dear Reviewer,

We are very grateful to your comments for the manuscript. According with your comments and suggestions, we amended the relevant part in manuscript. All of the comments and suggestions are responded as follows. All the lines and pages mentioned in responses are form the revised manuscript in review mode (Word version).

Comments and Suggestions. The authors have crafted an intriguing piece of work that undoubtedly captures the interest of readers and holds the potential to significantly advance the field of research. However, prior to publication, it is essential for the authors to attentively address the following comments to ensure the manuscript meets the highest standards of clarity, precision, and completeness. The manuscript requires several enhancements for clarity and visual appeal.

Response: Thank you for your comments and suggestions. We have read your comments carefully and attentively revised relevant part according your advice. Thank you again for your careful and rigorous review of the article and your affirmation of our work. We will keep working hard in the future. Thank you!

Comment 1. Firstly, in the title, it is advisable to include the full name of "Li-monium aureum."

Response: Thank you for your comment and suggestion. We have revised the title in the manuscript (page 1, line 3) according your suggestion. Thank you!

Comment 2. In the Abstract section, the author should provide the full form of LAH, highlight key study findings, and underscore the significance of the work, thereby reducing the introduction's length.

Response: Thank you for your comment and suggestion. We have drastically revised the abstract in response to your suggestions (page 1, lines 15-41), and the full form of LAH has been provided in the abstract section (page 1, line 22). Thank you!

Comment 3. Address Line 162 by questioning the rationale behind selecting specific material parameters in the experimental design, especially when these values were screened in steps 2, 3, and 4.

Response: Thank you for your comment and suggestion. The factors that may affect the yield of total flavonoids of LAH during ultrasound-assisted extraction include extraction temperature, extraction time, extraction power, number of extraction times, material-solvent ratio and ethanol concentration in the solvent, etc. Through the comprehensive consideration of the actual experimental operating conditions, five factors of extraction temperature, extraction time, extraction power, material-solvent ratio and ethanol concentration in solvent, were selected for the single factor experiment in this study. Under the condition that other factors were fixed, the influence of single factor on the total flavonoids yield of LAH at 6 levels was tested, and the numerical range of this single factor when the total flavonoids yield of LAH was the highest was screened. The fixed values set for each factor were the concentration of ethanol in solvent of 70 %, material-solvent ratio of 1: 20 g/mL, extraction temperature of 50 °C, extraction time of 60 min, extraction power of 80 W, the number of extraction times of 3 times. When one of the factors was screened, the screened factor was the variable, and the other factors were fixed using the set values above. The preliminary experiment results showed that the extraction time of six levels had little effect on the extraction rate of total flavonoids from LAH. While the effects of the other four factors on the extraction rate of LAH total flavonoids varied significantly at different levels, therefore, extraction temperature, extraction power, material-solvent ratio, and ethanol concentration in the solvent were selected as the key parameters for optimization of LAH extraction in the final experiment. As shown in Figure 1 of the manuscript, the yield of total flavonoids of LAH increased and then decreased when the ethanol concentration was the variable. When the ethanol concentration reached 70%, the yield of total flavonoids of LAH peaked at this screening range. Therefore, three levels centered on the peak value were selected for subsequent experiments, i.e., ethanol concentration of 60% - 80% in the extraction solvent. Similarly, the other three factors were screened according to this criterion. Thank you!

Comment 4. The figures in the manuscript demand an enhancement in quality. The font size throughout all figures is challenging to discern, particularly the X and Y labels and fonts. A crucial improvement lies in making these elements more visibly clear and reader-friendly. For example, Figure 1 needs a revamp for better representation—increasing the size and pixel quality, with a particular focus on elevating the font size of the X and Y axes. Subfigure 1's axes are challenging to follow, demanding an improvement. In subsequent figures, there is a consistent need for increased axis label font sizes to enhance legibility. Additionally, Figure 4 requires attention, as the current font size impedes clear visibility.

Response: Thank you for pointing this out. We apologize for the flaws in the quality of the figures. We have revised the unclear figures in the manuscript according your advice. Thank you!

Comment 5. Considering LAH is a mixture of flavonoids, the author should include the percentage composition of each component rather than relying solely on quercetin for quantification. A broader spectrum, as indicated by a calibration plot covering various components, would enhance the credibility of the study.

Response: Thank you for your comment and suggestion. There are some limitations to our study as you mentioned. These include the fact that the LAH extract, as a mixture, the specific component of its pharmacological activity is unclear. In this study, we used quasi-targeted metabolomics analysis to characterize and quantify the components of LAH extract, which resulted in the identification of a total of 1,666 components in LAH, and their specific information and relative quantitative values are shown in the supplementary file (Table S2). Although quasi-targeted metabolomics analysis provided some information on the components contained in LAH, more in-depth studies are still required to determine the specific components and their percentages in which LAH exerts pharmacological activities. We will utilize experimental techniques and methods of separation, purification, spectroscopy and chromatography in our future studies to explore in depth the specific information of the components contained in LAH in order to improve the quality and level of our research. Thank you again for your valuable comments, which has greatly helped to improve our research level. Thank you!

Reviewer 4 Report

Comments and Suggestions for Authors

The manuscript "Study on the alleviating effect and potential mechanism of Limonium aureum on LPS-induced inflammatory response in macrophage" focuses on  the anti-inflammatory properties of a natural product LAH and its mechanism of action, with the goal of developing it as a potential new natural anti-inflammatory drug to address the limitations of existing therapeutic options for inflammatory disease. The work is rather broad covering extraction optimization, selection of active compounds and activity studies.

There are several findings: 1) optimized ultrasound-assisted extraction of LAH,  2) LAH may have anti-inflammatory effects by modulating cytokine secretion; 3) application of quasi-targeted metabolomics to identify 96 active ingredients in LAH; 4)LAH could inhibit the protein expression of AKT1, RELA, and PTGS2 in macrophages, suggesting that LAH may regulate pro-inflammatory mediators through the AKT1/RELA/PTGS2 signaling pathway and alleviate LPS-induced inflammation in macrophages; 5) LAH's protective effect against LPS-induced inflammatory injury in macrophages may be achieved by inhibiting the production of pro-inflammatory cytokines through the regulation of the MAPK3/JUN signaling pathway.

The biological activity were performed in vitro only.

Overall, the work is well performed but rather difficult to follow especially in the discussion section. My suggestion is to move parts of the discussion into results section where each part is clearly defined and rearange as introduction-methodology-result-discussion in each part. Then discussion itself should be shortened and summarize only the major findings.

other comments:

l.121 - the term "OB, DL values" is used without prior explanation in the introduction (the explanation is in discussion instead)

Table 3. For average value only 3 digits are justifable (0,378) and 2 for RSD

(3,3%)

Figure 3 and 5 - poor quality, explain * and #

Author Response

Response to Reviewer 4:

Dear Reviewer,

Thank you very much for your careful and comprehensive review of our manuscript. We have revised the relevant part in manuscript according to your comment and advice. The responses answering each question are as follows. All the lines and pages mentioned in responses are form the revised manuscript in review mode (Word version).

Comments and Suggestions.

The manuscript "Study on the alleviating effect and potential mechanism of Limonium aureum on LPS-induced inflammatory response in macrophage" focuses on the anti-inflammatory properties of a natural product LAH and its mechanism of action, with the goal of developing it as a potential new natural anti-inflammatory drug to address the limitations of existing therapeutic options for inflammatory disease. The work is rather broad covering extraction optimization, selection of active compounds and activity studies.

Response: Thank you for your comments. Your recognition of our work gives us great encouragement. Thank you!

Comment 1. There are several findings: 1) optimized ultrasound-assisted extraction of LAH, 2) LAH may have anti-inflammatory effects by modulating cytokine secretion; 3) application of quasi-targeted metabolomics to identify 96 active ingredients in LAH; 4)LAH could inhibit the protein expression of AKT1, RELA, and PTGS2 in macrophages, suggesting that LAH may regulate pro-inflammatory mediators through the AKT1/RELA/PTGS2 signaling pathway and alleviate LPS-induced inflammation in macrophages; 5) LAH's protective effect against LPS-induced inflammatory injury in macrophages may be achieved by inhibiting the production of pro-inflammatory cytokines through the regulation of the MAPK3/JUN signaling pathway.

Response: Thank you for the comprehensive summary of our work. Thank you!

Comment 2. The biological activity were performed in vitro only.

Response: Thank you for pointing this out. The present study has some limitations, one of which is the study of biological activity conducted only in vitro. In this study, we demonstrated that LAH has good in vitro anti-inflammatory activity by establishing the LPS-induced inflammatory macrophage model. In the subsequent work, we will further investigate whether LAH has in vivo anti-inflammatory activity based on the results of the in vitro experiments and through the application of experimental animals. Thank you!

Comment 3. Overall, the work is well performed but rather difficult to follow especially in the discussion section. My suggestion is to move parts of the discussion into results section where each part is clearly defined and rearange as introduction-methodology-result-discussion in each part. Then discussion itself should be shortened and summarize only the major findings.

Response: Thank you for your comment and suggestion. We apologize for the inadequacies in the writing of the results and discussion sections. According to your suggestions, we have rearranged and substantially revised the results and discussion sections in order to better present and discuss our research results. Thank you for your valuable comments, which are essential to improve the quality of our articles. Thank you!

Comment 4. l.121 - the term "OB, DL values" is used without prior explanation in the introduction (the explanation is in discussion instead).

Response: Thank you for comment and suggestion. Due to the lack of clarity in the last paragraph of the Introduction, we have revised the paragraph in its entirety. The sentence containing the term "OB, DL values" has been removed from this section and the definitions and explanations of them would be provided in subsequent sections of the article. Thank you!

Comment 5. Table 3. For average value only 3 digits are justifable (0,378) and 2 for RSD (3,3%).

Response: Thank you for comment and suggestion. We are sorry for this carelessness. We have revised it in the manuscript (page 13, line 436) according your advice. Thank you!

Comment 6. Figure 3 and 5 - poor quality, explain * and #.

Response: Thank you for comment and suggestion. We are sorry for our negligence. We have redrawn Figures 3 and 5, and improved their quality. The meanings of * and # have been added in the annotations of Figure 3 and 5 in the manuscript (page 17-18, lines 529-531; page 25, lines 681-683). Thank you!

Round 2

Reviewer 2 Report

Comments and Suggestions for Authors

The authors have made considerable revisions to the manuscript. It can be recommended for publication.

Author Response

Response to Reviewer 2 (Round 2):

Dear Reviewer,

Thank you again for your comments and suggestion concerning our manuscript which help us greatly in improving the quality of our manuscript. The responses answering every comment are as follows. All the lines and pages mentioned in responses are form the revised manuscript in review mode (Word version).

Comments and Suggestions. The authors have made considerable revisions to the manuscript. It can be recommended for publication.

Response: Thank you for recognizing our revisions to the manuscript. Your help in the manuscript revision process has allowed our work to be better presented in the manuscript. Thank you again for your affirmation. Thank you!

Reviewer 3 Report

Comments and Suggestions for Authors

The authors have skillfully addressed the recommended revisions, leading to a noteworthy improvement in the manuscript's quality. I highly recommend proceeding with publication after a minor modification. It is suggested to spell out 'Lipopolysaccharide' (LPS) in full within the title for enhanced clarity

Author Response

Response to Reviewer 3 (Round 2):

Dear Reviewer,

Thank you again for your careful and comprehensive review of our manuscript. Your recognition of our work gives us great encouragement. All of the comments and suggestions are responded as follows. All the lines and pages mentioned in responses are form the revised manuscript in review mode (Word version).

Comments and Suggestions. The authors have skillfully addressed the recommended revisions, leading to a noteworthy improvement in the manuscript's quality. I highly recommend proceeding with publication after a minor modification. It is suggested to spell out 'Lipopolysaccharide' (LPS) in full within the title for enhanced clarity.

Response: Thank you again for your comments and suggestions on our manuscripts. It is your pertinent comments and constructive suggestions that have greatly improved the quality of our manuscript. We have revised the title of the manuscript according your suggestion (page 1, lines 3-4) to enhance the clarity. Thank you again for recognizing our revisions to the manuscript. Thank you!
